# Jigsaw++: Imagining Complete Shape Priors for Object Reassembly

## Abstract

The automatic assembly problem has attracted increasing interest due to its complex challenges that involve 3D representation. This paper introduces Jigsaw++, a novel generative method designed to tackle the multifaceted challenges of reconstructing complete shape for the reassembly problem. Existing approach focusing primarily on piecewise information for both part and fracture assembly, often overlooking the integration of complete object prior. Jigsaw++ distinguishes itself by learning a category-agnostic shape prior of complete objects. It employs the proposed "retargeting" strategy that effectively leverages the output of any existing assembly method to generate complete shape reconstructions. This capability allows it to function orthogonally to the current methods. Through extensive evaluations on Breaking Bad dataset and PartNet, Jigsaw++ has demonstrated its effectiveness, reducing reconstruction errors and enhancing the precision of shape reconstruction, which sets a new direction for future reassembly model developments.

## 1 Introduction

The challenge of object reassembly spans numerous applications from digital archaeology to robotic furniture assembly, and even to the medical field with fractured bone restoration. Object reassembly problems are classified into part assembly, which deals with semantically significant parts (Zhan et al., 2020; Schor et al., 2019; Li et al., 2020; Wu et al., 2020; Dubrovina et al., 2019), and fractured assembly, which handles pieces broken by substantial forces (Huang et al., 2006; Lu et al., 2023; Wu et al., 2023b). However, existing approaches face a critical limitation: they lack comprehensive understanding of the complete object when working with fragmentary inputs. This limitation is particularly acute in real-world scenarios where only a subset of fragments is available, and current reconstruction methods heavily rely on category-specific templates, c.f. (Thuswaldner et al., 2009; Papaioannou et al., 2017). It underscores the need for a new approach that could address these gaps and provide a complete shape prior to future research.

To address this fundamental challenge, we introduce Jigsaw++, a novel framework that bridges the gap between partially assembled pieces and the complete object prior. Rather than replacing existing assembly algorithms, our approach learns to synthesize plausible complete shape priors that can guide the reassembly process. While previous methods have attempted to compose shape priors (Yin et al., 2011; Zhang et al., 2015; Deng et al., 2023) , they typically impose restrictive constraints, such as requiring specific object categories or pre-existing complete shape templates. In contrast, Jigsaw++ learns to generate complete shape directly from partial assemblies, which enables our method to support a broader range of assembly scenarios.

Our approach draws inspiration from the recent success of 3D shape generators employing diffusion models, which map Gaussian noise to instances on the data manifold. Based on this principle, we propose to learn a complete shape prior through the generative model, then optimize the mapping from the partially assembled input towards this complete shape space. Ideally, this method will provide a realistic representation of what the complete object would look like based on the given input. Among many 3D representations, we focus on the point-cloud representation, due to its tight connection to the data acquisition devices and problem settings (Lu et al., 2023; Zhan et al., 2020).

Learning a point cloud generative model for fractured object reassembly is difficult. Most approaches require a fixed number of points and are also restricted to specific categories or need class conditioning. Another challenge is the scale of training data for learning shape priors. We overcome these challenges

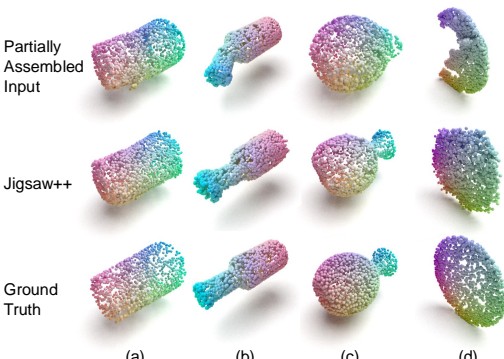

Partially
Assembled
Input

Jigsaw++

Ground
Truth

(a)          (b)          (c)          (d)

Figure 1: Overview of the problem setting. The input consists of a partially assembled object represented as a point cloud. The task requires the method to reconstruct a complete object from this input. We identify several representative challenges: (a) When the object is nearly fully assembled, the output should maintain the overall shape. (b) Although all parts are visible and present, their positions are misaligned. The algorithm needs to adjust their positions correctly. (c, d) In cases where parts are incomplete or significantly misplaced, the method should not only complete the object but also correct the displacements.

by adopting the LEAP image-to-3D reconstruction model (Jiang et al., 2024) under the point cloud representation. Our goal is to leverage its training on broad 2D datasets (Oquab et al., 2023). by developing a suitable mapping between raw point clouds and RGB images.

Drawing insights from contemporaty image editing approaches (Song et al., 2021a; Mokady et al., 2022; Meng et al., 2022), our model interprets the partially assembled object as user input, with the target being the complete object. This setup helps to utilize the learned shape generative model to predict the complete object from partial inputs. However, the difference in our setting is that the input is inaccurate and incomplete. Naively conditioning the output on the input still leads to inaccurate output. To address this issue, we introduce a "retargeting" phase which fine-tunes the mapping from the encoding of inaccurate input to the complete object output. This fine-tuning step significantly improves reconstruction quality.

In summary, our main contributions are as follows.

- We introduce Jigsaw++, a novel method that imagine the complete shape prior through retargeted rectified flow. The method generates comprehensive complete objects to serve as guides for the assembly process.
- We develop an object-level point cloud generation module capable of adapting to a large or arbitrary size of input and output point numbers. This model leverages the image-to-3D model and encompasses a joint generation of global embeddings and reconstruction latent via the rectified flow technique.
- The proposal of a "retargeting" strategy that links the reconstruction challenges in reassembly tasks with guided generation processes. This strategy facilitates the reconstruction of complete objects from partially assembled inputs and takes advantage of the straightness provided by rectified flow, resulting in lower tuning costs and higher flexibility.
- Jigsaw++ is orthogonal to the existing object reassemble methods. Our experiments on both the Breaking Bad dataset and PartNet demonstrate its adaptability to various assembly challenges and its ability to achieve significant improvements over baseline inputs.

## 2 RELATED WORK

### 2.1 OBJECT REASSEMBLY

Object reassembly problem falls into two primary categories: part assembly and fractured assembly. In part assembly, semantic-aware learning methods have emerged in recent years. Specific tools designed for the assembly of CAD mechanics have been developed (Jones et al., 2021; Willis et al., 2022). For the assembly of categorical everyday objects, research efforts (Schor et al., 2019; Li et al., 2020; Wu et al., 2020; Dubrovina et al., 2019) have concentrated on generating missing parts based on an accumulated shape prior to completing the entire object, although this approach can lead to shape distortions relative to the input parts. More recent works (Zhan et al., 2020; Harish et al., 2022; Li et al., 2023; Du et al., 2024) learns the part positions directly through regression or generative methods. However, these methods require the input objects to be semantically decomposed in a consistent manner and necessitate specific training for each object category.

The fractured assembly problem specifically addresses objects broken by extreme external forces. Previous research in this area typically falls into two categories: assembly based on fracture surface features or complete shape template. The former approach focuses on detecting fractured surfaces and extracting robust descriptors, with early work (Ruiz-Correa et al., 2001; Gelfand et al., 2005; Salti et al., 2014; Huang et al., 2006) employing hand-crafted features for assembly. More recent learning-based techniques have introduced methods (Chen et al., 2022; Wu et al., 2023b; Lu et al., 2023; Scarpellini et al., 2024) using learned features for matching local geometries, or predicting or generating piece positions. Another significant limitation of existing approaches is that they require that most of the fragments be available as input. However, this assumption is violated in real settings where a significant potion of fragments is missing (Thuswaldner et al., 2009; Papaioannou et al., 2017), in which prior knowledge of the complete object is critical.

Existing approaches that use information of complete shapes are template-based methods (Yin et al., 2011; Zhang et al., 2015; Deng et al., 2023). However, they often assume a specific complete shape for assembly, but are typically constrained by specific categories or challenges in generating accurate shape priors. Such settings do not apply to general-purpose fracture object reassembly.

## 2.2 3D OBJECT GENERATION

The field of 3D shape generation has witnessed significant progress, driven by the application of various generative models that produce high-quality point clouds and meshes. Techniques such as variational autoencoders (Yang et al., 2018; Gadelha et al., 2018; Kim et al., 2021) and generative adversarial networks (GANs) (Valsesia et al., 2018; Achlioptas et al., 2017) have been widely implemented to process 3D data. Further enhancements have been achieved through the integration of normalizing flows and diffusion models, which have spurred the development of state-of-the-art approaches (Yang et al., 2019; Kim et al., 2020; Zhou et al., 2021; Luo & Hu, 2021; Zeng et al., 2022; Lyu et al., 2023; Wu et al., 2023a; Mo et al., 2023; Zhang et al., 2023a; Gao et al., 2022). People also studied using 2D images and implicit neural fields to create text-guided 3D shapes (Xu et al., 2023; Ruiz et al., 2023; Lin et al., 2023; Cheng et al., 2023). Some approaches (Zhou et al., 2021; Lyu et al., 2021) also explored the generative shape completion which is highly relative to our task. These techniques strive to generate point clouds, SDFs, and meshes with both high fidelity and diversity, with some employing latent-based generation to even support multimodal 3D generation.

Our approach adopts comparable results in this space and addresses two fundamental challenges in point cloud generation. The first challenge is limited paired 3D data we have for learning a shape prior. Our approach develops a mapping between point clouds and RGB images, allowing us to use pretrained models that take 2D images as the input. The second challenge is point clouds with varying number of points. We again address this issue using the mapping between RGB images and point clouds, which enable us to generate 3D point clouds with many more points than prior approaches.

## 2.3 DIFFUSION MODEL AND RECTIFIED FLOW

Our approach uses state-of-the-art diffusion-based techniques for learning the shape prior and the mapping from inaccurate input to complete object output. Diffusion models (Ho et al., 2020; Song et al., 2021a; Dhariwal & Nichol, 2021; Zhang et al., 2023b; Podell et al., 2023; Song et al., 2021b) have demonstrated their versatility and effectiveness in a variety of generative tasks, including image, audio, and video generation (Saharia et al., 2022; Kong et al., 2020; Ho et al., 2022). These models operate via a forward process that incrementally adds Gaussian noise, coupled with a reverse process that gradually restores the original data, thus achieving high fidelity in the generated outputs. Beyond stochastic differential equation (SDE)-based approaches (Song et al., 2021b;a), recent efforts have emerged (Liu et al., 2023; Liu, 2022; Lipman et al., 2022; Albergo et al., 2023) focusing on directly learning probability flow ordinary differential equations (ODEs) between two distributions. This shift has led to improvements in generative efficiency and quality. Specifically, the introduction of Rectified Flow (Liu et al., 2023; Liu, 2022) implements a reflow process that significantly speeds up the generation process, which is effective in large-scale image generation (Esser et al., 2024; Liu et al., 2024). These collective advances highlight the transformative impact of diffusion models in various generative modeling tasks. This work focuses on developing a fractured object reassembly approach that uses these generative models under novel 2D-3D representations.

## 3 PROBLEM STATEMENT AND APPROACH OVERVIEW

We begin with the problem statement of Jigsaw++ in Section 3.1. Section 3.2 then presents an overview of Jigsaw++.

### 3.1 PROBLEM STATEMENT

Denote a collection of $n$ pieces as $\mathcal{P} = \{P_1, P_2, \cdots, P_n\}$, represented as point clouds of the surface of each piece. An assembly algorithm (e.g., Zhan et al. (2020); Lu et al. (2023)) produces a set of 6-DoF poses $\{T_1, T_2, \cdots, T_n\}$. These poses, derived from existing methods, partially restore the underlying object $\hat{O} = T_1(P_1) \cup T_2(P_2) \cup \cdots \cup T_n(P_n)$, where $T_i(\cdot), 1 \leq i \leq n$ is an operator that applies the transformation $T_i$ to piece $P_i$. The objective is to infer a possible set of complete 3D shapes $\mathcal{S} = \{S_1, S_2, \cdots, S_k\}$ based on $\hat{O}$ that share a similar outer shape with the original object $O$. Importantly, we aim for a data-driven approach where the complete restorations may contain geometries not present in the input. Fig. 1 provides a comprehensive overview of this problem.

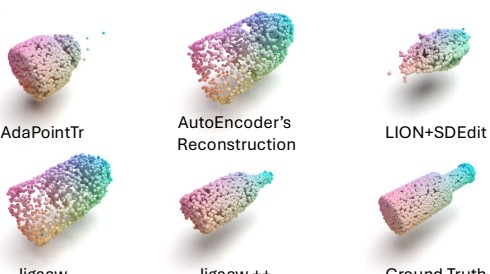

AdaPointTr    AutoEncoder's Reconstruction    LION+SDEdit

Jigsaw    Jigsaw ++    Ground Truth

Figure 2: Intuitive methods, including point cloud completion method AdaPoinTr (Yu et al., 2021), LION (Zeng et al., 2022) VAE's reconstruction, and editing method SDEdit (Meng et al., 2022), fails in providing shape prior when given partially assembled object.

To clearly establish the scope of this problem, we elucidate the following key aspects: (1) The **input** is the partially assembled objects from a prior algorithm, represented as point clouds. The state of this partially assembled object is not provided. There is no quantification of whether a piece is correctly assembled or how accurate the assembling is. (2) The **output** is a complete shape prior in point cloud form. This prior is not required to exactly replicate the geometric details of the input pieces, aligning with the template shape used in previous works (Yin et al., 2011; Zhang et al., 2015; Deng et al., 2023). However, a more accurate representation of the outer shape is preferred, as reflected in our evaluation metrics. (3) The **purpose** of this method is not to design a reassembly algorithm, but rather an additional layer of information to improve the reassembly algorithm. (4) Given the absence of prior work addressing this specific problem, we demonstrate how intuitive solutions fail in Fig. 2, highlighting the problem's **difficulty** and **uniqueness**. A detailed analysis of these results is presented in Appendix A.

### 3.2 APPROACH OVERVIEW

Jigsaws proceeds in two stages. The first stage learns a generative model to capture the shape space of complete objects. The second stage focuses on "regargeting" which reconstructs the complete shape from partially assembled inputs. Below we highlight the main characteristics of each stage.

**Learning Complete Shape Priors.** The first stage learns a generative model of point clouds that capture shape prior of the underlying objects. There are many available point cloud generative models (Zhou et al., 2021; Zeng et al., 2022; Lyu et al., 2023). However, there are two fundamental challenges in adopting them for our setting. First, most point cloud generative models are category specific and use a fixed number of points. Therefore, it is difficult to adopt them to learn a category agnostic model that requires different numbers of points capture geometric details of different categories of objects. Second, 3D data is sparse, which is insufficient to learn a category agnostic model to encode the shape space of objects in diverse categories.

Jigsaw++ adopts LEAP (Jiang et al., 2024), a pretrained multi-image-2-3D model to learn shape priors. LEAP uses DINOv2 features, which are trained from massive image data. In doing so, our generative model uses not only 3D data, but also 2D large-scale data. We introduce a bidirectional mapping between uncolored point clouds and RGB images. This mapping addresses the domain gap between raw 3D geometry and colored inputs to LEAP (as well as many other image-based 3D reconstruction model). It also nicely addresses the issue of having a limited number of 3D points. We will discuss details in Sec. 4.

**Reconstruction through Retargeting.** The second stage learns the reconstruction model that takes the assembly result of an off-the-shelf method as input and outputs a complete 3D model. A standard approach is to formulate this procedure as inversion-based methods (Song et al., 2021a; Mokady et al., 2022; Meng et al., 2022; Liu et al., 2023). In the image generation setting, the input is first inverted or mixed with noise and then re-generated.

The difference in our setting is that the inputs are biased partially assembled objects, and we do not have quantification of which part of the input is correct and which is not. In contrast, image-based conditions in existing approaches are unbiased complete objects. Due to this distribution shift, if we naively condition the learned generative model on the biased inputs, the resulting 3D shape is also biased. This is because not all latent codes in standard latent spaces correspond to valid 3D shapes. Addressing this issue requires a "retargeting" phase where the model is fine-tuned to understand the disparities between the partially assembled and complete objects.

In addition to fine-tuning, the typical approach for guidance-based generation in diffusion models involves performing reverse sampling, mixing the latent representation with a certain level of noise, and then executing forward sampling (standard generation). As diffusion-based models often require extensive sampling steps, we opt for the rectified flow (Liu et al., 2023) formulation, which allows for skip-over of steps during inverse sampling, thereby accelerating the fine-tuning process. This necessitates the use of rectified flow as the formulation for our generative model in the first stage. We will discuss details in Sec. 5.

## 4 GENERATION ON IMAGES-TO-3D

This section presents details on how to build a rectified flow based generation model for point cloud generation using an image-2-3D mapping. The generation pipeline is presented in Fig. 3.

**Bi-directional Mapping between Point Clouds and Images** Our generative framework is built upon a bi-directional mapping between point clouds and 2D images. Specifically, consider a normalized point cloud represented as $\boldsymbol{o} \in [0,1]^{N \times 3}$. Each point $\boldsymbol{o}_i \in [0,1]^3$ within this cloud, is associated with a function $f : [0,1]^3 \to [0,255]^3_{\mathbb{Z}}$. This function maps each point $\boldsymbol{o}_i$ to a color vector $\boldsymbol{c}_i \in [0,255]^3_{\mathbb{Z}}$ in the RGB space, where the mapping process is described by $\boldsymbol{c}_i = f(\boldsymbol{o}_i) = \lfloor 255\boldsymbol{c}_i \rfloor$. Please note that, although the color space is treated with integer values in this context, for applications involving image-to-3D reconstruction models, the color values can be maintained as fractional, thereby preserving accuracy throughout the transformation process. While similar coordinate-to-color mappings have been explored in pose estimation and reconstruction tasks (Wang et al., 2019; Sridhar et al., 2019), our work presents its first application to 3D generation.

The forward mapping from point cloud to image space is achieved through rasterization under specified camera poses. Conversely, the inverse mapping $f'$ reconstructs 3D coordinates from color values as $\boldsymbol{o}_i = f'(\boldsymbol{c}_i) = \frac{1}{255}\boldsymbol{c}_i$. This enables the recovery of point clouds from colored images encoded under our scheme. We further refine the reconstructed points through camera-ray alignment, projecting each decoded 3D point onto the ray connecting its corresponding pixel to the camera center. By aggregating multiple views from strategically selected camera poses, we can reconstruct a complete object point cloud with controllable point density.

This bi-directional mapping establishes a cyclic relationship: given a set of camera poses, we can render a sequence of images from a colored point cloud, and conversely, reconstruct the original point cloud from these images and camera parameters with high fidelity.

**A category agnostic image encoder.** The point could to image map described above opens the door to employ rich results in multi-view to 3D reconstruction models. Such models are trained from massive datasets. Some of them, including LEAP (Jiang et al., 2024), use the pretrained DINOv2 (Caron et al., 2021; Oquab et al., 2023) feature extractor, which boosts generalizability to novel categories. Jigsaw++ uses LEAP as the image encoder backbone. It provides a global embedding $\boldsymbol{g}$ from the input images and a reconstruction latent $\boldsymbol{r}$ for 3D reconstruction, we harness these global embeddings as the desired global latent for our generation model, aiming to simultaneously generate both the global and the reconstruction latents. Although only the latent reconstruction is directly utilized in the decoding phase, the global latent is generated throughout to help the model grasp global information of the input, which is vital for complete object reconstruction for object reassembly.

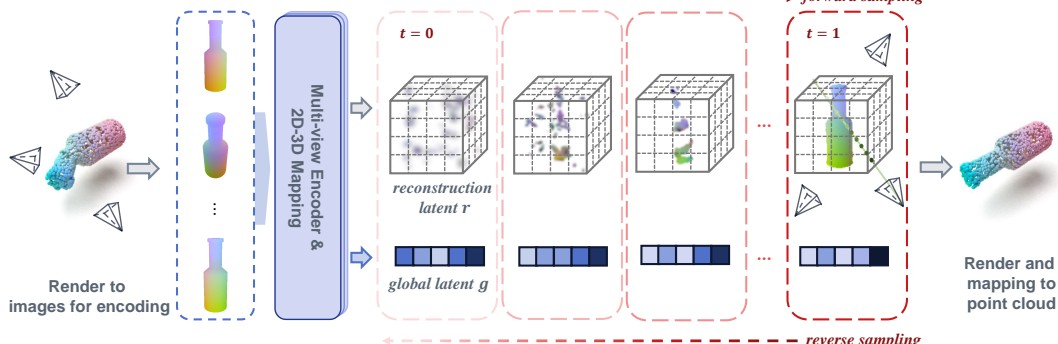

Figure 3: Generation on image-to-3D. The point cloud (or mesh if presented) is first rendered under specific camera parameters by mapping positions to RGB space. The image-to-3D reconstruction model then encodes these rendered images into both a reconstruction latent $r$ (here shows the decoded version of $r$) and a global latent $g$. A rectified flow model is trained to jointly generate these latents. Subsequently, the generated latents are decoded, rendered, and mapped back to a point cloud.

**Rectified Flow Generation.** Rectified Flow, as outlined in (Liu et al., 2023; Lipman et al., 2022), presents a unified ODE-based framework for generative modeling, facilitating the learning of transport mappings $T$ between two distributions, $\pi_0$ and $\pi_1$. In our images-to-3D model, $\pi_0$ typically represents a standard Gaussian distribution, while $\pi_1$ corresponds to the latent output of the image encoder.

The method involves an ordinary differential equation (ODE) to transform $\pi_0$ to $\pi_1$:

$$\frac{dZ_t}{dt} = v(Z_t, t), \text{ initialized from } Z_0 \sim \pi_0 \text{ to final state } Z_1 \sim \pi_1, \tag{1}$$

where $v : \mathbb{R}^d \times [0, 1] \to \mathbb{R}^d$ represents the velocity field. This field is learned by minimizing the objective:

$$\mathbb{E}_{(X_0, X_1) \sim \pi_0 \times \pi_1} \left[ \int_0^1 \left\| \frac{d}{dt} X_t - v(X_t, t) \right\| dt \right], \tag{2}$$

where $X_t = \phi(X_0, X_1, t)$ is an arbitrary time-differentiable interpolation between $X_0$ and $X_1$. The rectified flow specifically suggests a simplified setting where

$$X_t = (1 - t)X_0 + tX_1 \implies \frac{d}{dt} X_t = X_1 - X_0, \tag{3}$$

and the solver

$$Z_{t + \frac{1}{N}} = Z_t + \frac{1}{N} v(Z_t, t), \forall t \in \{0, \dots, N - 1\} / N. \tag{4}$$

This linear interpolation facilitates straight trajectories, promoting fast generation, as discussed in (Liu et al., 2024).

Rectified Flow offers two significant advantages: (1) it avoids assuming a fixed distribution for $\pi_1$, thus providing more flexibility in integrating the reconstruction encoder's learned distribution; (2) the model's ability to learn linear trajectories expedites both the forward and reverse sampling processes, benefiting the fine-tuning phase outlined in Sec. 5.

**Pipeline.** Given a set of 3D objects, our generator learns to generate objects that match the data space of the provided shapes through a three-stage process. In the *encode* stage, the colored 3D objects are rendered into images following camera settings from Kubric-ShapeNet (Greff et al., 2022). These images are then fed into DINOv2 (Oquab et al., 2023) and passed through a 2D-3D mapping layer both pre-trained using LEAP (Jiang et al., 2024), resulting in two types of latents: a voxel-based reconstruction latent $r$ and a global latent $g$ containing categorical information. The *generation* stage follows, where a joint latent rectified flow model is trained on the encoded latents. During inference, two latents are jointly generated as described in Eq. 4. The final stage, *decode*, involves converting the generated reconstruction latent $r$ into a neural volume. This neural volume is then rendered and converted into a point cloud, which represents the output of the entire pipeline.

To effectively handle the joint generation of the global and reconstruction latents, we employ the U-ViT (Bao et al., 2022) framework as our generative backbone. This structure has proven its

efficacy in image generation tasks (Bao et al., 2023; Esser et al., 2024), affirms its suitability for our application.

## 5 COMPLETE OBJECT RECONSTRUCTION

This section presents the details of the Jigsaw++ reconstruction module. We take inspiration from relevant approaches in image generation which transform user guidance into realistic outputs (Song et al., 2021a; Meng et al., 2022; Mokady et al., 2022; Liu et al., 2023). A common theme begins with inverse sampling based on given guidance, followed by forward sampling (generation) to produce the desired image in the target space.

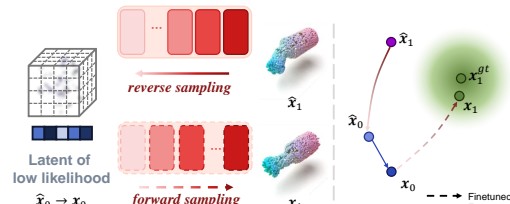

Figure 4: Reconstruction and retargeting. The reconstruction involves a reverse sampling stage to convert input to a latent. The latent will be perturbed to generate a complete shape. The retargeting is to provide guidance for those latent of low likelihood in $\mathcal{N}(0, I)$.

In the context of the reassembly problem, the partially assembled pieces using an off-the-shelf approach serve as the user-provided guidance Challenges, however, arise as previously discussed in Sec. 3. Unlike the 2D case where inputs are assumed accurate, our scenario demands larger adaptations, such as positional adjustments or the handling of non-observable overlapping pieces. These extensive modifications necessitate a targeted fine-tuning stage, which we term "retargeting".

Given the partially assembled object $\hat{O}$ and its associated latent $\hat{x}_1 = (\hat{g}_1, \hat{r}_1)$ (representing a set of global and reconstruction latents), we can employ a reverse ODE solver to determine the latent $\hat{x}_0$. Since the input is not a naturally assembled complete object, $\hat{x}_0$ is likely to have low likelihood under $\pi_0 = \mathcal{N}(0, I)$. To adjust this, we apply Langevin dynamics:

$$\boldsymbol{x}_0 = \alpha\hat{\boldsymbol{x}}_0 + \sqrt{1 - \alpha^2}\xi, \ \xi \sim \mathcal{N}(0, I), \tag{5}$$

which moves it to a region of higher likelihood.

Ideally, a subsequent forward sampling from $\boldsymbol{x}_0$ should yield a $\boldsymbol{x}_1$ that accurately represents the learned complete shape space. However, given the significant discrepancies between the input partially assembled object and the target, we find that fine-tuning with data pairs $(\boldsymbol{x}_0, \boldsymbol{x}_1)$ is necessary to more effectively guide our generative model. The objective for this stage is,

$$\mathbb{E}_{\boldsymbol{x}_0, \boldsymbol{x}_1}\|(\boldsymbol{x}_0 - \boldsymbol{x}_1) - v(\boldsymbol{x}_t, t)\|^2, \tag{6}$$

where $\boldsymbol{x}_0$ is computed as Eq. 5 and $\boldsymbol{x}_1$ corresponds to the ground truth of the complete object.

We again use rectified flow (Liu et al., 2023; Liu, 2022) to train this reconstruction module. The efficiency and straightness of the rectified flow is critical; they enable a substantial reduction in the number of steps required during the reverse sampling phase - to just $1/25$ of the original steps - while preserving a faithful latent representation. This efficiency is key to decreasing the fine-tuning cost.

## 6 EXPERIMENT AND EVALUATION

### 6.1 EXPERIMENT SETUP

**Dataset.** We use the Breaking Bad dataset (Sellán et al., 2022) for the fracture assembly problem. The Breaking Bad Dataset encompasses a diverse array of synthetic physically broken patterns for the task of fracture assembly problem. Our experiments were conducted on the everyday subset of this dataset, consisting of 498 models with 41,754 distinct fracture patterns. This subset is segmented into a training set with 34,075 fracture patterns from 407 objects, and a testing set containing 7,679 fracture patterns from 91 objects. The average diameter of the objects in both the training and testing sets is 0.8. The generative model is trained only on the training set to ensure a fair comparison. Categorical information is not provided during the experiments.

For the part assembly problem, we employed PartNet (Mo et al., 2019), following the approach of previous work DGL (Zhan et al., 2020) for training and evaluation. PartNet offers a large collection of

Table 1: Quantitative results of baseline methods and Jigsaw++ on the Breaking Bad dataset and ParNet. Jigsaw++ consistently improves performance of the baseline method across all settings.

| Breaking Bad Dataset | | | |
|---|---|---|---|
| Method | CD ($\times 10^{-3}$) $\downarrow$ | Precision (%) $\uparrow$ | Recall (%) $\uparrow$ |
| SE(3) (Wu et al., 2023b) | 22.4 | 20.2 | 22.5 |
| w/ Jigsaw++ | 14.3 | 37.8 | 36.6 |
| Difference | -8.1 | +17.6 | +14.1 |
| Jigsaw (Lu et al., 2023) | $10.5 \pm 0.1$ | $45.6 \pm 0.1$ | $42.7 \pm 0.1$ |
| w/ Jigsaw++ | $4.5 \pm 0.3$ | $48.7 \pm 0.2$ | $49.5 \pm 0.3$ |
| Difference | -6.0 | +3.1 | +6.8 |

| PartNet | | | | | | | | | |
|---|---|---|---|---|---|---|---|---|---|
| | Chair | | | Table | | | Lamp | | |
| Method | CD | Pre. | Rec. | CD | Pre. | Rec. | CD | Pre. | Rec. |
| DGL (Zhan et al., 2020) | 47.8 | 21.5 | 20.0 | 53.6 | 16.6 | 15.4 | 68.8 | 18.6 | 17.9 |
| w/ Jigsaw++ | 41.0 | 52.0 | 33.6 | 42.6 | 53.6 | 31.0 | 46.3 | 42.3 | 28.5 |
| Difference | -6.8 | +30.5 | +13.6 | -11.0 | +37.0 | +15.6 | -22.5 | +23.7 | +10.6 |

daily objects with detailed and hierarchical part information. We selected the same three categories as prior work: 6,323 chairs, 8,218 tables, and 2,207 lamps, adhering to the standard train/validation/test splits with the finest level of segmentation used. We independently trained the model on three subsets, ensuring that the validation/test sets were not included in the training set of the generation model.

**Metrics.** We adopted two types of evaluation metrics to evaluate the performance of our proposed methods. (1) *Shape difference*. The chamfer distance defined by $CD(S1, S2) = \frac{1}{|S_1|} \sum_{x \in S_1} \min_{y \in S_2} \|x - y\|_2^2 + \frac{1}{|S_2|} \sum_{y \in S_2} \min_{x \in S_2} \|x - y\|_2^2$, is used to assess the differences between the ground truth shape, the partially assembled shape, and the reconstructed global shape. (2) *Shape accuracy*. We follow a similar idea of F-score to define the precision and recall metric as precision $= \frac{1}{|S_{gt}|} \sum_{x \in S_{gt}} \mathbf{1}_{\mathsf{Dis}(x, \mathsf{NN}(x,S)) \leq \eta}$, and recall $= \frac{1}{|S|} \sum_{x \in S} \mathbf{1}_{\mathsf{Dis}(x, \mathsf{NN}(x,S_{gt})) \leq \eta}$, to evaluate how closely the reconstructed shape matches the ground truth. Here, $\mathsf{Dis}(\cdot)$ is a distance function, and $\mathsf{NN}(\cdot, \cdot)$ is to find the nearest neighbor of one point in another shape.

**Baseline Methods.** We compare our methods with state-of-the-art assembly algorithms for the fracture and part assembly problem: SE(3) (Wu et al., 2023b), Jigsaw (Lu et al., 2023) and DGL (Zhan et al., 2020). All methods are open-source with available model checkpoints, which we used to generate the partially assembled inputs for our model and comparison. Since our algorithm works orthogonally to existing methods, it is sufficient to demonstrate its superiority by demonstrating improvements over these methods.

## 6.2 PERFORMANCE

**Overall Performance.** We evaluated the performance of baseline methods with our proposed Jigsaw++ on both Breaking Bad dataset (Sellán et al., 2022) for the fracture assembly problem and PartNet (Mo et al., 2019) for the part assembly problem. A quantitative analysis is detailed in Table 1.

Jigsaw++ consistently outperformed the baseline methods, demonstrating its capability to reconstruct a meaningful underlying complete shape that corresponds closely to the input partially assembled objects. Even with a less favorable initialization algorithm SE(3) (Wu et al., 2023b), our algorithm can give a large improvement on their results. Specifically, Jigsaw++ achieves significantly better results in terms of reconstruction error in the fracture assembly problem. We draw three insights: (1) The original size of the objects in the Breaking Bad Dataset is considerably smaller compared to those in PartNet (please refer to Sec. 6.3 for a failed reconstruction case on PartNet). This small size discrepancy enables the mapping between point clouds and images to pose minimal impacts on the representation of the complete shape. (2) The diversity of complete shapes in the Breaking Bad Dataset is less varied than in PartNet, simplifying the modeling of the complete shape space.

Despite less favorable initialization in part assembly, Jigsaw++ significantly improves the precision and recall metrics to depict complete shapes on PartNet. Since the assembled object from DGL could be significantly displaced or reordered, Jigsaw++ offers valuable insights into the likely overall

Table 2: **Left**: Reconstruction performance of Jigsaw++ when presented with input with missing pieces. The model are tested on the Bottle category of the Breaking Bad dataset. **Right**: Fracture assembly performance with original-shape matching with the shape prior generated by Jigsaw++.

| Breaking Bad - Bottle | | | | | Breaking Bad | | | | |
|---|---|---|---|---|---|---|---|---|---|
| Method | Input | CD↓ $\times 10^{-3}$ | Precision↑ % | Recall↑ % | Method | Matching Type | MAE(R)↓ degree | MAE(T)↓ $\times 10^{-2}$ | PA↑ % |
| Jigsaw | complete | 3.4 | 52.8 | 49.9 | Jigsaw | fracture | 36.3 | 8.7 | 57.3 |
| Jigsaw++ | complete | 1.8 | 61.0 | 59.4 | Jigsaw++ | + GT shape prior | 17.8 | 3.6 | 73.1 |
| Jigsaw++ | 20% missing | 2.0 | 59.5 | 59.4 | Jigsaw++ | + 20% noise shape prior | 18.2 | 3.7 | 72.6 |

shape. Such insight on the complete shape is essential for the general object reassembly problem, and provides a new possibility for developing better algorithms for the object reassembly problem.

**Performance with Missing Pieces.** To demonstrate the effectiveness and the robustness of the proposed method, we conduct a test using the Bottle category from the Breaking Bad dataset. Each piece will have 20% probability of been removed and we ensure at least one piece is presented in one object. We input the Jigsaw's result with pieces removed to the Jigsaw++ model.

As shown in Table 2 left, the resilience of Jigsaw++ is evidenced when processing inputs with 20% missing pieces. Under these conditions, the model maintained a low CD of $2.0 \times 10^{-2}$, with precision and recall approximately at 59.4%. This performance closely aligns with that seen in fully intact inputs, highlighting Jigsaw++'s robustness in dealing with data incompleteness.

**Performance for Fracture Reassembly Algorithm.** While our primary interest lies in applying the generated shapes to reassembly algorithms, we encountered challenges in finding an algorithm that effectively utilizes the complete shape prior. To demonstrate the potential of our approach in assembly problems, we present an alternative evaluation method.

We augment the Jigsaw algorithm (Lu et al., 2023) by providing a matching between the original object surface and our generated shape prior during its global alignment stage. This matching is computed by finding the closest point from the ground truth position of each point to the generated shape. It is important to note that the fractured surface matching and global alignment algorithm remain unchanged from Jigsaw and may contain errors.

Table 2 right shows that when using the closest point matching with ground truth, we can reduce Jigsaw's error by 50%. Even with the introduction of 20% noise to this "ground truth" matching, performance remains significantly improved over the baseline Jigsaw algorithm. These results demonstrate that our generated shape can indeed assist assembly algorithms. This suggests that future research efforts to develop algorithms that can fully utilize these complete shape priors could yield significant advancements in reassembly tasks.

**Ablation Study on varying Parameters.** We now show how different parameter settings influence the performance during the "retargeting" phase of Jigsaw++. We first examine the effect of the rectified flow formulation under varying reverse sampling steps. As discussed in Sec. 5, this formulation significantly reduces the required number of reverse sampling steps. Letting $N$ denote the forward sampling steps, and $N_r = kN$ the reverse sampling steps, we explore the effects of altering $k$ on reconstruction outcomes. The results, illustrated in the upper row of Fig. 5, show that the model performs best when $k = 1/10$. A full reverse sampling phase tends to overly mimic the input, which is suboptimal for reconstruction. Moreover, setting $k$ too low can cause the latent to deviate excessively, leading to a different output.

Further, we explored the impact of modifying the latent composition $\boldsymbol{x}_0 = \alpha \hat{\boldsymbol{x}}_0 + \sqrt{1 - \alpha^2}\xi, \xi \sim \mathcal{N}(0, I)$ on reconstruction quality. Research in image generation, such as those by (Liu et al., 2023; Meng et al., 2022), indicates that a larger $\alpha$ generally replicates the input more closely, while a smaller $\alpha$ pushes the generation towards the data domain. We observed a similar trend in our generative model as in Fig. 5 lower row. At $\alpha = 1$, the output is very similar to the input, whereas decreasing $\alpha$ makes the result progressively diverge towards representing a complete object. Interestingly, although the precise shape might not be replicated, the reconstructed form invariably aligns visually with the ground truth category.

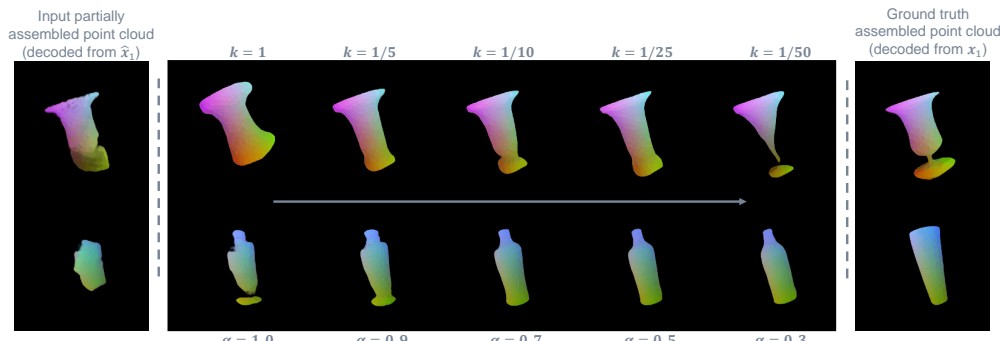

Figure 5: Ablation study of Jigsaw++ with varying parameters on the Breaking Bad dataset. Top: Varies the reverse sampling steps to $N_r = kN$ to assess how well the rectified flow model accommodates step reductions. Bottom: Alter the $\alpha$ parameter in the Langevin dynamics to explore how changes in latent resampling during the retargeting phase affect model performance.

## 6.3 LIMITATION AND FAILURE CASES

While we have investigated various strategies to enhance the robustness of point cloud generation, our model still struggles to generalize to unseen object types or significantly varied objects. We identify three main types of failure cases as in Fig. 6: (a) Size limitation in color mapping. Converting object point clouds into color spaces imposes significant size constraints. Objects like tall street lights might not be adequately visible in the rendered images, causing the reconstruction process to fail. Conversely, the model tends to perform better with smaller objects. (b) Dataset limitations. Given that our model is trained on selected datasets, it struggles

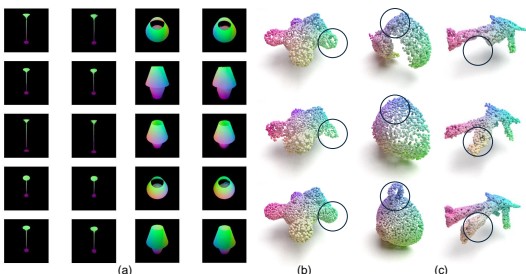

Figure 6: Three types of failure cases of Jigsaw++. (a) Size limitation in color mapping. (b) Limitation on unseen objects. (c) Topology constraints.

to recognize and reconstruct rarely encountered or unseen object types. Specific details cannot be accurately reconstructed using the current methodology. Larger datasets are in need for adapting to more complex scenarios which we leave for future work. (c) Topological and Geometrical Accuracy: The model exhibits limitations in preserving complex topological structures, particularly when reconstructing objects with intricate geometric features. For example, when processing images of mugs where the handle is partially occluded or ambiguous in the input, the model successfully reconstructs the main body but struggles to accurately reproduce the handle geometry and its connectivity. The generative process occasionally introduces spurious artifacts that deviate from the ground truth geometry, a limitation inherent to the current probabilistic formulation of the reconstruction problem. While our approach improves upon existing methods, the outlined limitations underscore the necessity for employing larger and better models, as well as richer datasets, in future research efforts to address these challenges.

## 7 CONCLUSIONS AND FUTURE WORK

In this study, we present Jigsaw++, a novel framework developed to tackle the challenge of complete shape reconstruction in object reassembly tasks. Jigsaw++ utilizes a novel point cloud generative model that reimagines the complete object shape from partially assembled inputs. By incorporating image-to-3D reconstruction techniques, Jigsaw++ adeptly navigates the challenges of scale and diversity in training data. Additionally, we show the rectified flow formulation enhances our proposed "retargeting" phase, establishing a more robust connection between the latent space and the complete object space. Experimental results demonstrate Jigsaw++'s superior reconstruction performance, marking a significant improvement over existing methods. Although we have achieved successful reconstructions, we have yet to devise methods to effectively leverage our outputs as guidance for further reconstructions. This limitation opens up new avenues for research in the field of object reassembly.

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

# APPENDIX

## A   POTENTIAL SOLUTIONS AND HOW THEY WORKS

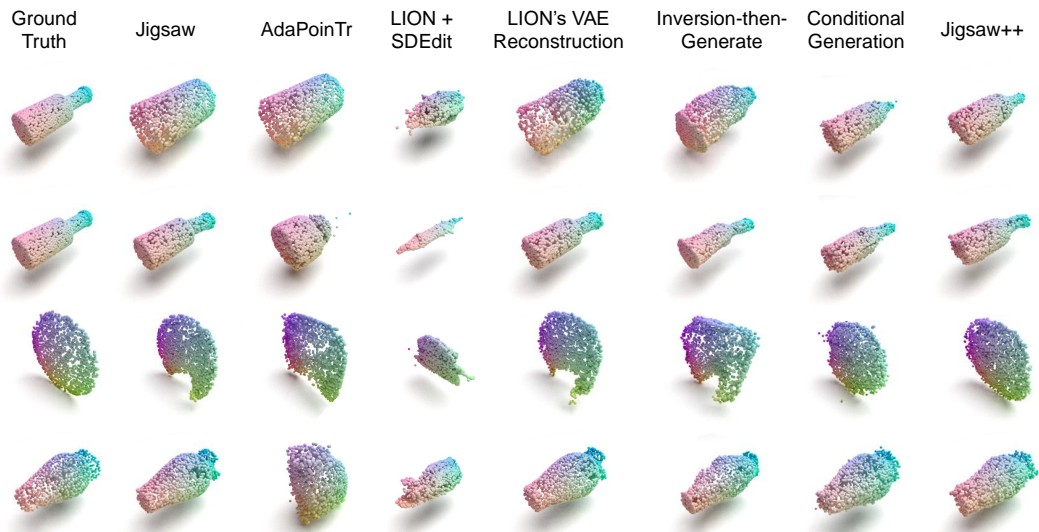

Figure 7: Qualitative demonstration of potential solutions.

Generating a complete shape prior based on a partially assembled object is a relatively new problem that is often underestimated in its complexity. We explored several intuitive solutions during the development of our method to demonstrate the challenges involved. While it is impossible to enumerate all potential solutions, we have selected representative approaches to highlight the uniqueness of our task. The difficulty in providing an accurate shape prior stems from two main challenges: (1) Lack of quantification of assembly errors: We do not know which pieces are correctly assembled and which are not. (2) Balancing shape alteration: The algorithm must adapt to varying degrees of assembly accuracy, from minor adjustments for nearly perfect assemblies to significant corrections for misplaced or incomplete pieces.

We tested four representative algorithms using state-of-the-art models and evaluated their performance on four test cases. Figure 7 illustrates the results of these experiments.

**Point Cloud Completion**   We adopted AdaPoinTr (Yu et al., 2023) using their open-sourced code and model trained on the ShapeNet dataset. We provided the algorithm with a subset of correctly placed pieces from Jigsaw's result. The algorithm exhibited the following limitations: (a) It interpreted the subset of parts as a complete shape, resulting in no additional completion as in the first bottle. (b) With more parts in the second bottle, it completed the top slightly, but the was sparse and limited in range. (c) It produce a resonable result for the plate which most closely resembled a typical completion task, while for the vase, it over-correct the given input.

**Point Cloud Generative Model with Editing**   We employed LION (Zeng et al., 2022) with the SDEdit (Meng et al., 2022) model using their open-sourced code. The results showed that (a) the generated shapes with similar overall forms (e.g., thin long shape for bottle input, flat shape for plate), but (b) unable to to consistently maintain the correct object category.

**Point Cloud Auto-encoders**   We utilized LION's (Zeng et al., 2022) VAE to assess the effectiveness of reconstruction. Results showed that the output was mostly identical to the input, with only minor changes towards the desired shape. This behavior is consistent with the VAE's objective of accurate shape reconstruction.

While these methods excel in their designed tasks, they fall short in addressing the specific challenges of inferring complete shape prior for the reassembly problems.

**Inversion-then-Generate**  We evaluate the effectiveness of direct inversion-then-generate pipeline to show how "retargeting step" influence the result. Using the same generator parameters and inversion settings as Jigsaw++ experiments, we observe that this baseline approach yields improvements on simpler cases (e.g., bottles and vases with minor variations). However, it demonstrates significant limitations when substantial modifications are required, exhibiting failure patterns similar to direct VAE reconstruction. These results suggest that the inversion-then-generate approach alone lacks the flexibility to accommodate major structural changes, underscoring the importance of our retargeting mechanism in handling complex shape difference.

**Conditional Generation**  One potential solution is to train a conditional generative model by finetuning our first-stage model with partially assembled inputs as conditions, similar to techniques used in recent 2D generation tasks (Ho et al., 2021; Zhu et al., 2023; Valevski et al., 2024). We implement this by incorporating partial assembly point clouds as additional input tokens during the finetuning process.

This conditional approach achieves stronger baseline performance with a Chamfer Distance of $4.8 \times 10^{-3}$, 46.3% precision, and 50.6% recall. While its Chamfer Distance matches our retargeting method, the precision falls below input level despite achieving higher recall. Qualitative analysis reveals the underlying behavior: the model excels at smoothing input geometry and completing missing regions (hence higher recall) but struggles to correct misplaced parts (resulting in lower precision). In contrast, our retargeting approach achieves a better balance among the three key challenges of this task: correcting misplaced parts, completing missing regions, and maintaining valid shape structure. This comparison validates the effectiveness of our retargeting strategy in handling the unique requirements of assembly-guided shape prior generation.

## B  IMPLEMENTATION DETAILS

### B.1  USED CODEBASES AND DATASETS

For baseline comparison, the following codes are used:

- DGL (Zhan et al., 2020): https://github.com/hyperplane-lab/Generative-3D-Part-Assembly.
- SE(3) (Wu et al., 2023b): https://github.com/crtie/Leveraging-SE-3-Equivariance-for-Learning-3D-Geometric-Shape-Assembly/tree/main.
- Jigsaw (Lu et al., 2023): https://github.com/Jiaxin-Lu/Jigsaw, (MIT License).
- PoinTr and AdaPoinTr (Yu et al., 2023; 2021): https://github.com/yuxumin/PoinTr, (MIT License).
- LION (Zeng et al., 2022): https://github.com/nv-tlabs/LION, (NVIDIA Source Code License).
- SDEdit (Meng et al., 2022): https://github.com/ermongroup/SDEdit, (MIT License).

For building our methods, the following codes are referenced:

- LEAP (Jiang et al., 2024): https://github.com/hwjiang1510/LEAP.
- UViT (Bao et al., 2022): https://github.com/baofff/U-ViT, (MIT License).
- Rectified Flow (Liu et al., 2023): https://github.com/gnobitab/RectifiedFlow.

The following datasets are used:

- Breaking Bad Dataset (Sellán et al., 2022): doi:10.5683/SP3/LZNPKB (License as listed in the link).
- PartNet (Mo et al., 2019): The [Pre-release v0] version at https://partnet.cs.stanford.edu/ for mesh, and the version presented with DGL (Zhan et al., 2020).
- Kubric-ShapeNet (Greff et al., 2022): The version with LEAP for camera parameters.

### B.2  PARAMETERS

We provide a detailed model parameters in Table. 3.

Table 3: The detailed experiment parameters.

| | Parameter | Breaking Bad Dataset | | PartNet | | description |
|---|---|---|---|---|---|---|
| | | base | retargeting | base | regargeting | |
| Training | epoch | 500 | 100 | 1000 | 400 | training epochs |
| | bs | 32 | 32 | 16 | 16 | batch size |
| | lr | 0.0001 | 0.00002 | 0.0001 | 0.00002 | learning rate |
| | optimizer | Adam | Adam | Adam | Adam | optimizer during training |
| | scheduler | Cosine | - | Cosine | - | learning rate scheduler |
| | min_lr | 1e-6 | - | 1e-6 | - | minimum learning rate for Cosine scheduler |
| | frames | 5 | 5 | 5 | 5 | input frames to the image encoder |
| Model | $N$ | 100 | 100 | 100 | 100 | sample steps in Rectified Flow |
| | $N_r$ | - | 4 | - | 4 | reverse sampling steps in retargting |
| | $\alpha$ | - | 0.5 | - | 0.5 | scaling factor for latents during retargeting |
| | depth | 12 | | 768 | | depth of UViT |
| | $d$ | 768 | | 768 | | token dimension in UViT |

## C  TRAINING DETAILS

### C.1  TRAINING RESOURCES AND INFERENCE TIME

Our experiments utilized a setup featuring eight NVIDIA Tesla A100 GPUs, with all running times based on this specific GPU configuration.

The finetuning of LEAP reconstruction model takes 232 GPU hours. In the training phase for the base generative model, different datasets required varying amounts of GPU time: the Breaking Bad dataset needed 480 GPU hours, while the PartNet categories required 40 GPU hours for Lamp, 216 GPU hours for Chair, and 240 GPU hours for Table. Additionally, the "retargeting" stage fine-tuning took 480 GPU hours for the Breaking Bad dataset and half of base model for each PartNet category.

For inference, reverse sampling of a single instance on one GPU took 0.2 seconds, and forward generation took 5 seconds. The complete processing time for one instance, includes rendering and reconstruction, was approximately 7.5 seconds on average.

### C.2  TRAINED MODELS

On the Breaking Bad dataset of the fracture assembly problem, LEAP (Jiang et al., 2024) is first finetuned using rendered mesh data. One generation model is trained for the entire subset without categorical information. Then, this model is finetuned and "retargeted" based on data computed by Jigsaw for the reconstruction task. The same model without finetuning on SE(3) (Wu et al., 2023b) is used for testing on SE(3) (Wu et al., 2023b) model.

On the PartNet of the part assembly problem, LEAP is first finetuned using rendered mesh data for all three categories. For each category, one generation model is trained, which results to three base generative models. These models are finetuned independently for the reconstruction task based on data computed by DGL.

### C.3  DATA VISUALIZATION

Figure 8 provides a visualization of the rendered data utilized in our experiments. In the top row, the partially assembled object is shown in point cloud format, which is used as input during the "retargeting" phase and for testing. The bottom row features the rendered complete objects, which are based on the mesh data from the dataset. Due to the superior continuity of this mesh data, it is selected as the ground truth for guiding the training of the generative model and the image-to-3D model, LEAP.

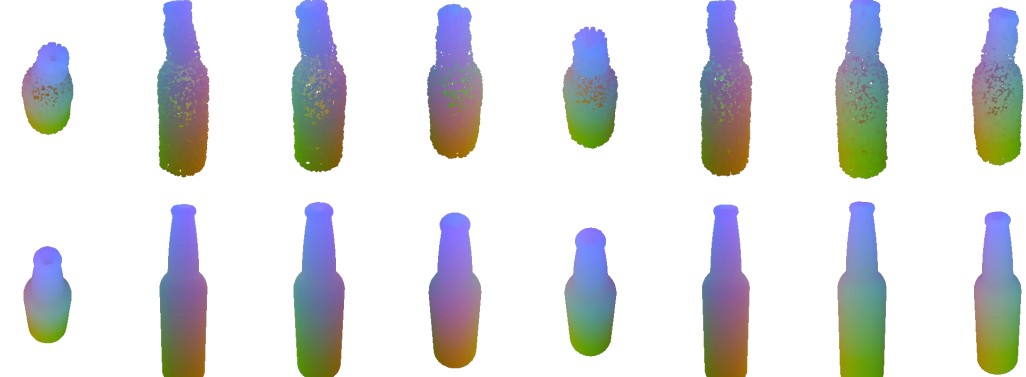

Figure 8: Visualization of one instance from the Breaking Bad Dataset. Top: The input partially assembled object is presented in point cloud format, which is employed both in the "retargeting" phase and for testing purposes. Bottom: Input complete objects rendered from meshes. Those data are used to create ground truth data for the training phases of the generative model and the image-to-3D model LEAP.

# D    ADDITIONAL RESULTS

## D.1    METRIC DISTRIBUTION

Fig. 9 presents a comprehensive analysis of metric distributions, illustrating the impact of our generative approach on reconstruction quality. While the incorporation of generative models introduces inherent uncertainties - manifesting as point displacement, omission and addition of geometric features, or even shape change - the quantitative improvements are substantial. Notably, Jigsaw++ demonstrates a remarkably lower peak in Chamfer distance distribution compared to baseline methods, with a larger proportion of samples achieving high precision and recall scores. These improvements in geometric accuracy, when considered alongside the assembly performance metrics (Table 2, Right), provide compelling evidence that the learned shape priors serve as a valuable constraint for the reassembly task. The integration of complete shape priors introduces an additional layer of geometric reasoning that effectively guides the reconstruction process, particularly in challenging cases.

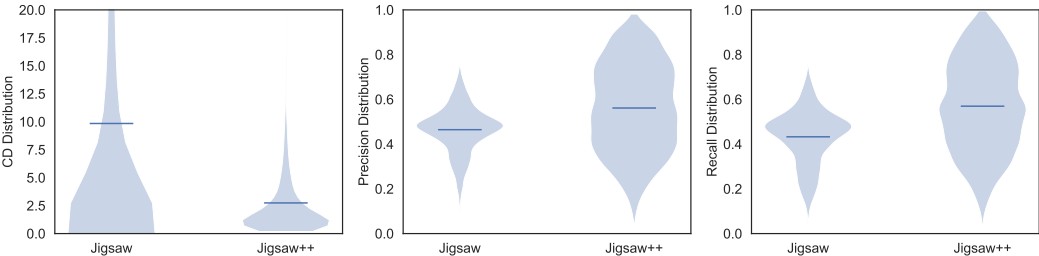

Figure 9: Metric distribution of Jigsaw and Jigsaw++. The metric Chamfer Distance is truncated by $[0.0, 20.0]$ for better visual quality.

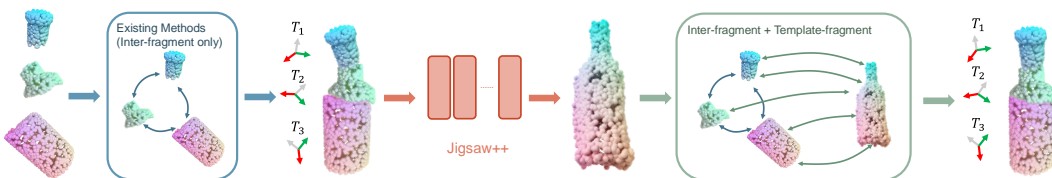

Figure 10: Apply Jigsaw++ to assembly workflow. Left: Use existing methods to compute an initial assembly result and compose a partially assembled object. Middle: Jigsaw++ generates a complete shape prior from this partial assembly, providing global context unavailable to local matching methods. Right: The shape prior guides refinement of fragment transformations through template matching.

# E    APPLY JIGSAW++ TO ASSEMBLY PROBLEM

To clarify how Jigsaw++ enhances practical assembly tasks, we present a complete pipeline overview in Fig. 10. Our method serves as an intermediate step that provides complete shape prior as an additional level of information to improve assembly accuracy.

Given fragment point clouds, the pipeline operates in three phases:

1. **Initial Assembly:** Existing methods (e.g., Jigsaw, SE(3)) compute initial fragment placements using local geometric features, producing a partially assembled object.

2. **Shape Prior Generation:** Jigsaw++ processes this partial assembly to generate a complete shape prior in the same point cloud representation as the input fragments.

3. **Assembly Refinement:** The shape prior guides fragment placement optimization (through geometric matching in our example) between fragments and the complete shape. This produces refined transformation matrices for each fragment, improving the final assembly accuracy.

Importantly, this pipeline can be extended in future work by developing more sophisticated matching algorithms between fragments and shape priors, or by incorporating the shape prior directly into existing assembly optimization objectives.

# F    VISUALIZATION

We present detailed visualization of results on the Breaking Bad Dataset (Fig. 12) and PartNet (Fig. 13). We also present a visualization on several examples we tested on Fantastic Breaks (Lamb et al., 2023) (Fig. 11). We use the same model trained on Breaking Bad Dataset. Please note that Fantastic Breaks only involves 2-pieces samples and all objects are real-world objects that doesn't exist in the Breaking Bad Dataset. For the fracture assembly problem, we additionally visualize the experiment described in Table 2 Right where we use this shape prior generated by Jigsaw++ to guide assembly. Each instances are organized in the order of "partial input - Jigsaw++ - Ground Truth" vertically.

# G    BROADER IMPACTS

This paper tackles object reassembly problem, which has no known negative impact on society as whole. On the contrary, its application in archaeology and medication would benefits research in other areas. Our method utilizes 3D generative model, which we hope could address several hard problems overlook by the current researches. The data we use are all objects datasets. Although we

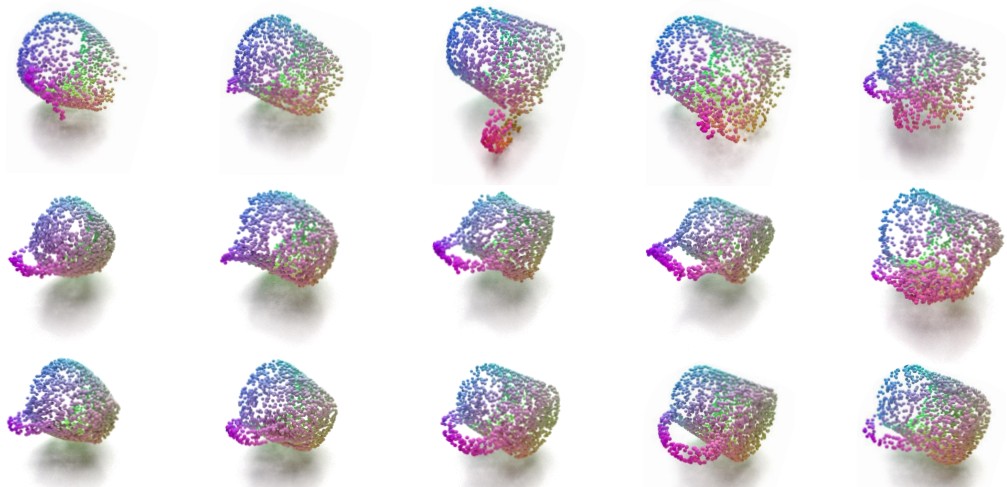

Figure 11: Detailed visualization of results on the FantasticBreaks.

see no immediate negative use cases or content from this model, we acknowledge the necessity of handling the generative model with care to prevent any potential harm.

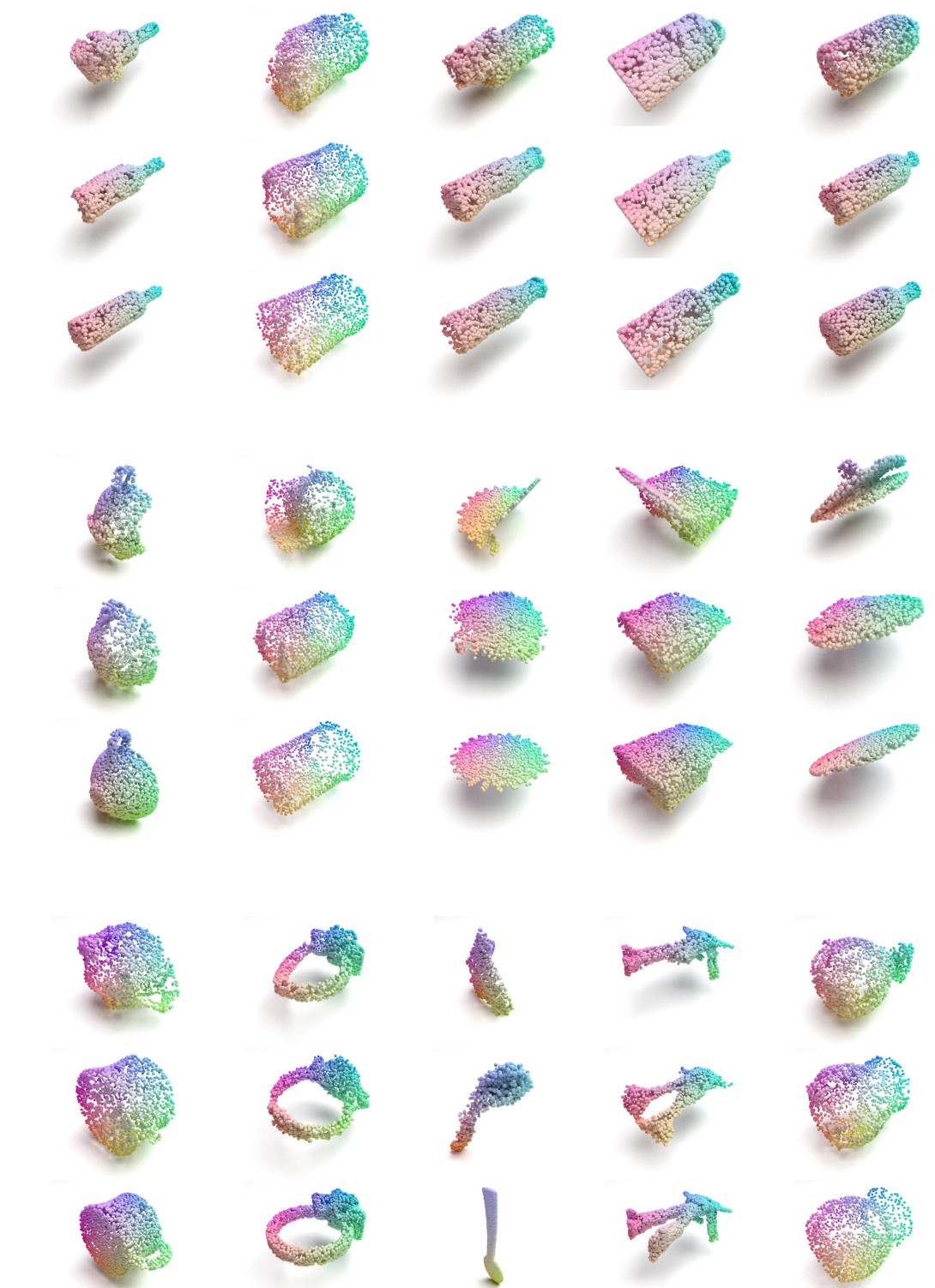

Figure 12: Detailed visualization of results on the Breaking Bad Dataset.

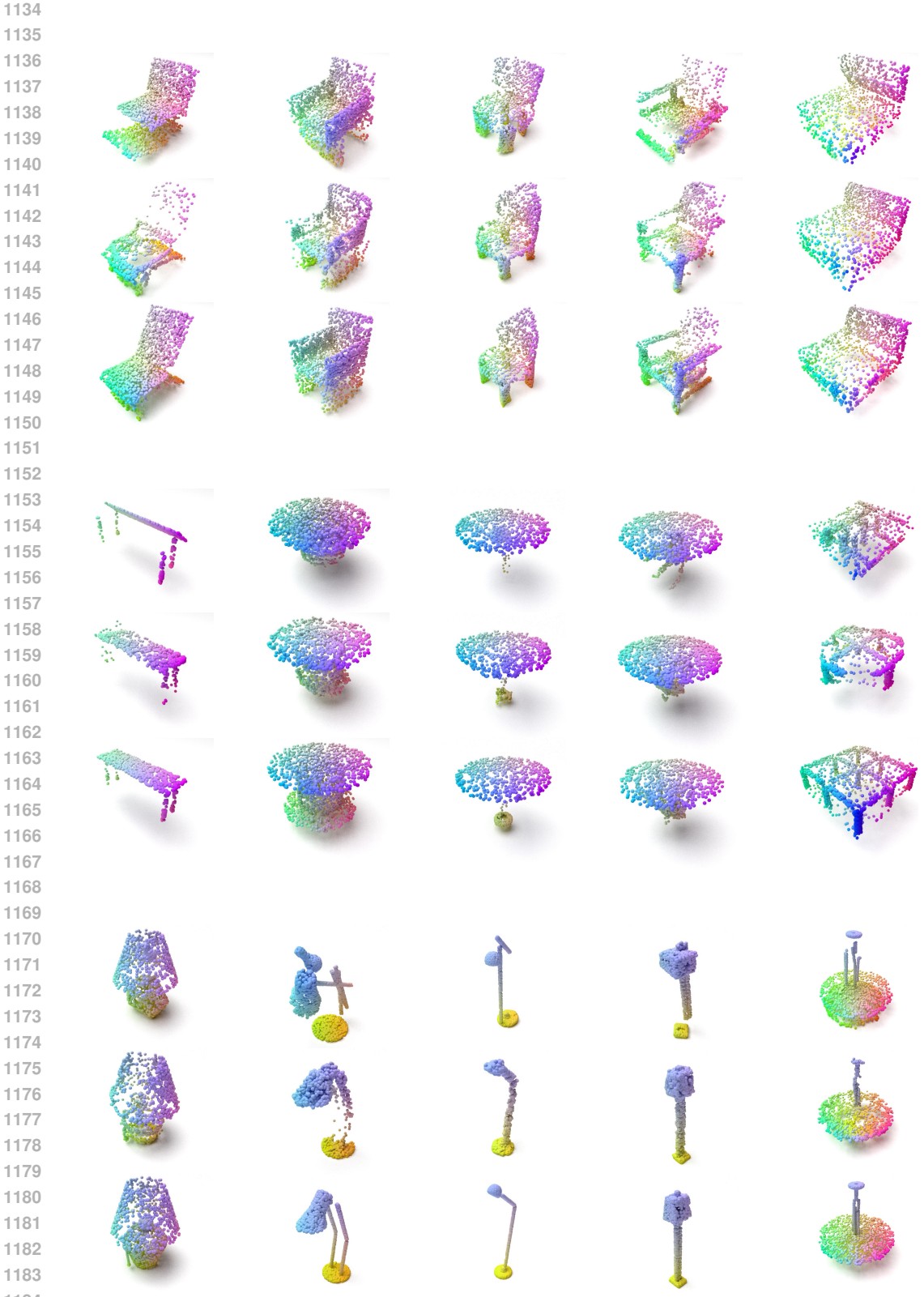

Figure 13: Detailed visualization of results on the PartNet.

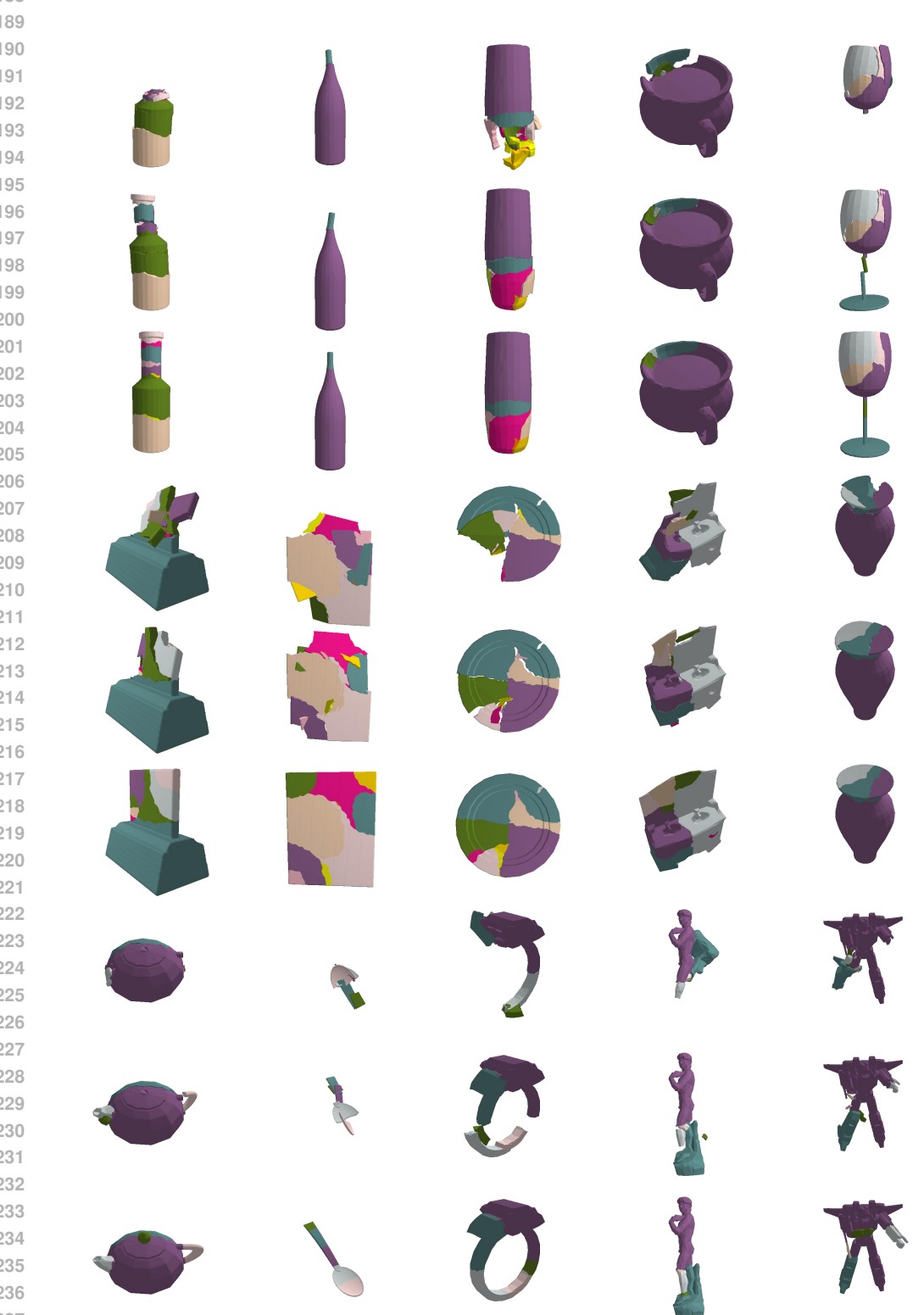

Figure 14: Visualization of fracture assembly performance with original-shape matching with the shape prior generated by Jigsaw++.