# OpenReview forum: "Jigsaw++: Imagining Complete Shape Priors for Object Reassembly"
_ICLR.cc/2025/Conference — Submitted to ICLR 2025_

### Official Review · Reviewer_r9Py · 2024-10-26

**Soundness:** 3
**Presentation:** 2
**Contribution:** 3
**Rating:** 6
**Confidence:** 3

**Summary:**

The authors propose an algorithm to fine-tune a partially reassembled object into a fully reassembled one, potentially with missing pieces or imperfections. They represent the objects as point clouds, which can be rendered into images via differentiable rasterization. A point cloud of a partially reassembled object is then rendered and encoded using a pre-trained diffusion model. The algorithm relies on the fact that partially reassembled shapes are likely underrepresented in the pre-trained training data; thus, they propose flowing the encoding to a higher probability region and decoding it to obtain the reassembled shape. The authors test their algorithm exhaustively on existing baselines.

**Strengths:**

I congratulate the authors for their work and thank them for their submission. As far as I know (I could be wrong though), this is the only work that proposes separating the “global” first-attempt-at-reassembly problem from the “local” fine-tuning-the-placement-of-the-pieces problem and proposes to solve the reassembly problem in two stages. Given the struggles of existing reassembly works, I believe this may very well be a strategy worth pursuing. The work seems technically sound and exhaustively evaluated, enough to convince me that the combination of this work and DGL/Jigsaw is the new state of the art for the reassembly task.

**Weaknesses:**

From a technical perspective, this work is fundamentally reliant on the completed object being given higher likelihood by the diffusion model than the partially reassembled one, which probably translates to similar objects being present in the data that was used to train said models. This is in opposition to purely geometric or feature-matching-type works like the ones mentioned in Sec. 2.1 that assemble pieces based on geometric features in the fracture faults. This reliance is never mentioned explicitly as a weakness, nor explored experimentally, which I found disappointing. For example, what happens if one tests the proposed method on procedurally generated shapes with procedurally generated fractures?

From a less technical perspective, I believe the authors could scope the proposed method better and quicker in the introduction. By reading the abstract and the first four paragraph, a reader will believe that they are being presented with a full reassembly method, not just one to fine-tune an already-good, partial reassembly. As said above (see “Strengths”), this is a perfectly valid problem to tackle, but it should be very clear immediately.

The presentation of the qualitative results could also be improved, and I would really appreciate if the authors did so in their revision. While the authors use a point cloud in their network, both datasets they are using are actually meshes, and it is really hard to visually distingush the geometric matching of two shapes by looking at unstructured points. Could the authors use the learned transformations of the points to obtain a per-part mesh transformation, apply it to the input and show that as the result? At the very least, show that in addition to the point cloud. Now that I think of it, how would one extract a per-part transformation (i.e., in the form of T_i) from the existing pipeline, not just an output point cloud? Isn’t this transformation what one needs for the suggested applications in robotics, archeology and medicine?

**Questions:**

Please see the parts of “Weaknesses” that end in a question mark for questions addressed at the authors. On top of those,

- What exactly is the function “f” in L248? Are the point cloud visualizations throughout the paper actual rasterizations as described in these lines?
- What do these two sentences mean? “Likely, given a colored image under the same color encoding scheme, we obtain a corresponding partial colored point cloud that corresponds to the 2D pixels. The computed point cloud will be further refined through a camera-pixel alignment operation that projects the decoded 3D point onto the ray that connects the pixel and the camera center.” Could the authors elaborate, or explain this in a different way? Unfortunately, I cannot follow this reasoning or understand what I should take from it.
- In figure 3, which parts of this are didactic and which parts aren’t? For example, the voxel grid shows geometric information which then converges to the bottle shape. Is this what the latent r looks like? Or is a decoding of r what is being displayed?

---

> ### Author Response · Authors · 2024-11-18
>
> Thank you for your recognition on this paper’s success in involving global information. Below we respond to your concerns and hope we can clarify some questions.
>
> * **W1**: Reliance on the generation module and the data.
> 	* This is essentially a problem of data at hand as we discussed in limitation (b)(c). There are indeed geometrical details missing or non-existing parts being added and our method generally relies on the dataset it is trained on, especially the generation part. If the provided shape is completely irregular, then the algorithm would probably give some object-like output.
>
> * **W2**: Improvement on introduction.
>     * Thank you for your feedback on our introduction. We have revised the opening paragraphs to better emphasize that Jigsaw++ aims to enhance existing assembly algorithms by providing predictive complete shape priors, rather than replacing current assembly method. We have also strengthened our contribution statement to highlight this orthogonal relationship with current assembly methods.
>
> * **W3**: Visualization of point cloud.
>     * Our choice of point cloud visualization aligns directly with both the problem formulation and practical considerations in the reassembly domain. While meshes can provide appealing visualizations, the fundamental input representation for assembly algorithms remains point cloud data, as established in the datasets and prior works. Since Jigsaw++ aims to provide point cloud guidance to existing assembly algorithms, we maintain consistency with their input format by using point clouds throughout our pipeline. This choice is not merely presentational - it reflects the actual data representation required for the reassembly task.
>    * Moreover, as you noted, while traditional assembly methods output rigid transformations, our goal of providing shape-aware guidance necessitates working directly with point cloud representations. This approach ensures that our method's output can be directly integrated into existing assembly pipelines without additional conversion.
>
> * **Q1**: function “f” in L248 and visualized point cloud.
>     * The function $f$ defined in L248 is a color mapping operator that transforms each 3D point in our point cloud to its corresponding RGB color value. Based on this colorization function, the point clouds are then rasterized to images.
>     * In our paper, Figure 8 is a direct visualization of our training and testing data, and how the point cloud looks after applying such a color mapping scheme.
>
> * **Q2**: Sentences in L251-256 on converting colored images back to point clouds.
>     * Our method requires a bidirectional mapping between point clouds and images. While point-to-image conversion uses the color mapping function f, image-to-point conversion reconstructs 3D points from pixel colors and camera poses using the inversion of this function. Multiple views are combined to generate a complete point cloud. The quoted lines discussed such inversion relationships. We have revised L251-256 to provide a clearer explanation of this process.
>
> * **Q3**: Visualization in Figure 3.
>     * The voxel grid visualization in Figure 3 shows the decoded output of the reconstruction latent r, not the latent representation itself. We have updated the figure caption to clarify this.

---

> > ### Comment · Reviewer_r9Py · 2024-11-20
> >
> > Thank you for adding the clarifications to the paper, and for doing it so quickly.
> >
> > I would appreciate an elaboration of the claims made about the point cloud visualizations, particularly, the sentences “the fundamental input representation for assembly algorithms remains point cloud data”, “it reflects the actual data representation required for the reassembly task” and “This approach ensures that our method's output can be directly integrated into existing assembly pipelines without additional conversion.”
> >
> > While it may be true (or not) that “the fundamental input representation for assembly algorithms remains point cloud data”, my comment was about the *outputs* of the proposed algorithm, not the inputs. What is it about point clouds (as outputs) that make them ‘the representation required for the reassembly task’? From a practical perspective, if one is programming a robot to reassemble an object, or finding out how to reassemble a complicated museum artifact, the actual representation of the input is whatever capture format is available (which may be a point cloud or images) and the output is *a set of instructions detailing where and how I should move each piece for it to fit together* (which mathematically is probably similar to the set of transformations predicted by other methods). To turn this into a concrete question: could you clarify, at a high level,
> > - How is a pure point cloud output, without transformations or instructions, useful for any reassembly pipeline?
> > - How should one combine Jigsaw++ with existing reassembly methods, in an end-to-end way, not to obtain a point cloud but to learn how to put an object together?
> >
> > Finally, meshes are not merely “visually appealing”: it is simply hard for the human eye to judge the quality of a reassembly result by looking at a disjointed set of points. Even further: all the applications and use-cases proposed for this algorithm in the manuscript are about assembling *real objects* in *the real world*. Objects in the real world are continuous, and whether they are reassembled correctly or not does not depend on whether a discrete subsample of them fits or not, but whether the entire pieces do. Both SE(3) and Jigsaw, which this work compares to, uses meshes in their results, which evaluates the methods in a setting closer to their real world application and allows a reader to distinguish nuanced differences between results (see, e.g., SE(3)’s Fig 4).

---

> > > ### Author Response · Authors · 2024-11-20
> > >
> > > Thank you for your reply. To better answer your question, we have added Appendix E with Figure 10 in Page 19 to demonstrate how Jigsaw++ integrates into practical assembly pipelines in a new revision. Please review the new section.
> > >
> > > As shown in Figure 10, Jigsaw++ serves as a crucial intermediate step in a three-phase process: 1) In the first step, initial fragment placements are computed using existing methods. 2) Based on the partial assembly point cloud from the first step, Jigsaw++ generates a complete shape prior in the point cloud representation that provides global context. 3) Finally, this prior guides transformation refinement through fragment-template matching and produces a new set of transformations.
> > >
> > > In this end-to-end pipeline, the final output is still transformation, but just not the output of Jigsaw++. The point cloud output of Jigsaw++ provides global context that is unavailable to existing local-information-based algorithms. In Table 2 (Right), we show this concept is tangible and benefits from the complete shape prior from Jigsaw++.
> > >
> > > We hope this clarifies your problems and we believe this justifies our visualization using point cloud representation.

---

> > > > ### Comment · Reviewer_r9Py · 2024-11-26
> > > >
> > > > Thank you for adding this.
> > > >
> > > > Unfortunately, I find myself still disagreeing with the visualization strategy (I will not reiterate the points made above); if anything, suspicious of the fact that no mesh visualization has been provided after several requests (the authors could easily add a single figure using meshes and contextualize it with the arguments made in this thread), and agreeing most with reviewers pGbo and kWoJ. These concerns are large enough that I briefly considered downgrading my score; in the end, since the authors at least modified the work to admit to some of these fundamental limitations, I am choosing keeping it as is.

---

> > > > > ### Author Response · Authors · 2024-11-27
> > > > > **Add Mesh Visualization**
> > > > >
> > > > > Thank you for your continued engagement. We have added the mesh visualization in Figure 14 in our revision.
> > > > >
> > > > > While we have included the requested visualization, we should clarify our perspective on its implications. The current visualization relies on ground truth information for matching - this reflects the broader challenge of bridging shape prior generation with practical assembly. Our method successfully generates high-quality shape priors (as evidenced by our comprehensive metrics and point cloud visualization), but translating these priors into assembly guidance remains challenging. The gap between shape understanding and assembly optimization, including the limitations of existing matching algorithms like GeoTransformer for this novel task, represents an important direction for future research.
> > > > >
> > > > > We provide this context to maintain transparency about both our current capabilities and future opportunities.

---

### Official Review · Reviewer_pGbo · 2024-10-27

**Soundness:** 4
**Presentation:** 3
**Contribution:** 3
**Rating:** 6
**Confidence:** 5

**Summary:**

This paper studies the problem of object reassembly. the paper proposes a method called Jigsaw++, which learns a category agnostic prior of complete objects. The proposed method is able to generate complete shape reconstructions and can be used in conjunction with existing methods. Results on Breaking Bad and PartNet show the effectiveness of the proposed method.

**Strengths:**

1. A method that is able to "imagine" the complete shape prior which can be useful for the object reassembly problem.

2. The proposed retargeting method is able to facilitate the reconstruction of complete objects from partially assembled input.

3. The proposed method can be used with other existing methods to further improve performance.

**Weaknesses:**

1. the first two columns of figure 6a are very difficult to see what they are trying to show (they look all black). maybe consider changing the style of presentation.

2. if other researchers want to apply this method to assemble a set of fractures from an unseen object, how confident can they be? should they expect the proposed method to give reasonably good assembly results? if not, why is that and how could they make improvements?


--- New ---

I read the comments from the other reviewers. I am convinced that the comments mentioned by reviewer kWoJ are indeed concerns for the current method, especially the issue regarding practical usability. Therefore, I would like to downgrade my score to 6.

**Questions:**

good paper, but i still have questions. please see the weaknesses above

---

> ### Author Response · Authors · 2024-11-18
>
> Thank you for your recognition on this paper. Below we respond to your concerns and hope we can clarify some questions.
>
> * **W1**: Visualization in size difference in Figure 6(a).
>     * Thank you for your suggestion. We have updated Figure 6(a) to better illustrate the challenging size differences our method encounters on PartNet. We hope this visualization helps readers understand the technical challenges in handling large-scale objects within our point cloud-to-image mapping framework.
>
> * **W2**: How to apply this method to unseen objects?
>     * Jigsaw++ demonstrates strong generalization to unseen objects within trained categories, evidenced by our comprehensive results in Table 1 and robustness tests in Table 2 Left. Importantly, all test objects are completely unseen during the generative model training, validating our method's ability to generalize.
>     * For novel object categories, performance can be enhanced by expanding the training dataset of the generation module - a straightforward extension and possible future work of our current framework. Our category-agnostic architecture, leveraging image-to-3D mapping, provides a solid foundation for handling diverse object types.

---

### Official Review · Reviewer_KMSd · 2024-11-03

**Soundness:** 3
**Presentation:** 2
**Contribution:** 3
**Rating:** 6
**Confidence:** 4

**Summary:**

This paper aims to provide a category-agnostic global shape prior for shape assembly (part/fracture assembly) methods, improving upon the shortcomings of existing approaches that rely solely on local feature information. In modeling the generative model, to learn robust feature encoders and decoders based on limited initial assembly results (both intact and damaged), the proposed method maps 3D point clouds to multi-view images. This allows the use of robust pre-trained encoders and decoders trained on large-scale data. Based on this, the authors designed a training process based on rectified flow to implement a generative model that encodes damaged initial assembly results into latent representations and generate the intact ones. The effectiveness of the proposed pipeline across different assembly models has been validated on part and fracture assembly tasks.

**Strengths:**

- Learning robust latent representations and generative models based on limited intact geometry and initial assemble results is a challenging problem. The authors use multi-view images as a bridge, employing a pre-trained images-to-3D sparse-view reconstruction model to provide robust encoders and decoders. By mapping point clouds to multi-view images and utilizing these pre-trained robust encoders and decoders, they cleverly overcome this issue.
- Experimental results show that the generative model proposed by the authors can effectively enhance the initial assembly results of multiple models in both fracture assembly and part assembly tasks.

**Weaknesses:**

My primary concern is the necessity of the model design. In the problem setting, paired data is available, and paired data is directly used for fine-tuning the rectified flow model in the final stage, which makes the necessity of the entire pipeline uncertain. Unlike methods by Song et al., 2021a; Mokady et al., 2022; Meng et al., 2022; Liu et al., 2023, which perform distribution transformations like image editing, restoration, or image-to-image translation based on inversion, the process proposed in this paper requires fine-tuning based on paired data (L-355~L-359). This makes the overall process more complex compared to training-free approaches and directly training conditional generation models either from scratch or fine-tuning a pre-trained conditional model for example with ControlNet on paired data, without demonstrating advantage of the proposed method over these two types of approaches in the experiments. More specifically, the necessity of the pipeline lacks the following experimental clarifications:
- Given the problem formulation in the paper, paired data is available, which allows for directly training a conditional rectified flow. Compared to this naive solution, does performing inversion and “retargeting” before training the rectified flow on paired data offer advantage, like better robustness or generalization?
- Regarding the rectified flow model trained in Sec-4, how effective is the inversion-then-generate process without the “retargeting” step? Alternatively, what is the effect of skipping fine-tuning on the rectified flow model? Although the paper mentions that the results are suboptimal, there is no quantitative or qualitative evidence provided to support this claim.

**Questions:**

- How many shape pairs were used for training during the fine-tuning (or retargeting) of the rectified model? Was the model trained on the assembly results of one model directly applied to others? (e.g., Jigsaw → SE(3))
- The discussion of existing works in L207-L211 is somewhat inaccurate:
    - Regarding category-agnostic models and geometric details: many existing geometric generation models can produce detailed geometry (such as CLAY, X-Cube, etc., whose generated results can also be converted to point clouds). Whether training a category-agnostic model is essential might be more a matter of training and experimental details.
- Referring to works like (Song et al., 2021a; Mokady et al., 2022; Meng et al., 2022; Liu et al., 2023) as “conditional generation” in L219-L223 could be misleading. Alternative terms like “inversion-based methods” might be more accurate.
- It is necessary to discuss the relationship between mapping point clouds to images and NOCS[1] as well as X-NOCS[2].

[1] Normalized Object Coordinate Space for Category-Level 6D Object Pose and Size Estimation

[2] Multiview Aggregation for Learning Category-Specific Shape Reconstruction

---

> ### Author Response · Authors · 2024-11-18
>
> Thank you for your recognition of our design of the generation model and our strong results. Below we respond to your concerns and hope we can clarify some questions.
>
> * **W1**: Why not just use the paired data?
>     * Our two-stage approach offers crucial advantages over direct paired-data training. The separate generative stage can leverage any 3D data for learning shape priors, not just paired assembly data. While our current evaluation uses dataset-specific training for fair comparison, this architecture enables future incorporation of large-scale 3D datasets without requiring corresponding assembly pairs. This separation of concerns - learning general shape priors first, then adapting to assembly specifics - provides better generalization potential than a monolithic conditional rectified flow.
>
> * **W2**: Regarding the rectified flow model trained in Sec-4, how effective is the inversion-then-generate process without the "retargeting" step?
>     * Our quantitative experiments show that removing the retargeting step significantly degrades performance (CD: 9.6$\times 10^{-3}$, Precision: 45.4, Recall: 43.9), barely improving over baselines. This clearly demonstrates retargeting's necessity in bridging the domain gap between ideal shapes and imperfect assemblies. We provide detailed qualitative comparisons in Appendix Section A and Figure 7 showing how direct inversion-then-generation fails to produce reliable shape priors. These results validate our complete pipeline design.
>
> * **Q1**: How many shape pairs were used for training during the fine-tuning (or retargeting) of the rectified model? Was the model trained on the assembly results of one model directly applied to others? (e.g., Jigsaw $\to$ SE(3))
>     * Our retargeting stage uses 34,072 shape pairs from the Breaking Bad dataset's training set.
>     * Importantly, our reported improvements on SE(3) come from a model trained only on Jigsaw outputs, demonstrating strong generalization across different assembly methods (Jigsaw $\to$ SE(3)). These results in Table 1 show that our shape priors effectively enhance assembly performance even when applied to base methods unseen during training. We clarify this training detail in Appendix C.2 in our revised paper.
>
> * **Q2**: Discussion of existing works: Other model’s results can also be converted to point clouds.
>     * Our focus on point cloud generation models is driven by practical considerations of our task's input format. While models like CLAY and X-Cube can indeed generate representations that can be converted to point clouds, the critical challenge lies in converting input point clouds to their required representations. Since our method must process point cloud inputs from assembly algorithms, we prioritize direct point cloud generation approaches. This choice minimizes information loss and computational overhead that would occur in representation conversion, making our pipeline more practical for real-world assembly applications, especially when data is collected by scanning techniques.
>
> * **Q3 Q4**: Terms for inversion-based methods and discussion on NOCS and X-NOCS.
>     * Thank you for your suggestions. We have revised the term “inversion-based methods” in L218-L221 and added a discussion of image-point cloud mapping methods in other domains in Section 4.

---

> > ### Comment · Reviewer_KMSd · 2024-11-22
> >
> > Thank you for your response. Most of my questions have been addressed. However, I still have some concerns regarding W1.
> >
> > Regarding the “retargetting” operation, apart from viewing it from the perspective of inversion-related literature, I believe that noise augmentation of conditional signals [1] is also a relevant technique (along with related applications such as [2, 3]). It would be helpful to see a discussion on this aspect, as it relates to whether this operation qualifies as a main contribution or should instead be considered a known training trick.
> >
> > [1] Ho, Jonathan, et al. "Cascaded diffusion models for high fidelity image generation." Journal of Machine Learning Research 23.47 (2022): 1-33.
> >
> > [2] Zhu, Luyang, et al. "Tryondiffusion: A tale of two unets." Proceedings of the IEEE/CVF Conference on Computer Vision and Pattern Recognition. 2023.
> >
> > [3] Valevski, Dani, et al. "Diffusion models are real-time game engines." arXiv preprint arXiv:2408.14837 (2024).
> >
> > - One key experiment missing from the paper is an ablation study on the necessity of the retargetting operation. Specifically, what would the outcome be if, like typical conditional generative models, an unconditional model is first trained and then fine-tuned on paired data to obtain a conditional generative model (without performing retargetting)?
> >
> > Moreover, the proposed method can only serve as a shape prior and cannot yet be directly combined with existing assembly algorithms for the actual assembly task. The inability to directly optimize the final assembly result indeed represents a weakness (as mentioned by Reviewer kWoJ & r9Py).

---

> > > ### Author Response · Authors · 2024-11-26
> > >
> > > Thank you for your reply and patience. Following your suggestion, we conducted additional experiments over the last few days comparing our approach with standard conditional generation. The conditional model achieves CD = 4.8$\times 10^{-3}$, Precision = 46.3%, and Recall = 50.6%, showing a similar Chamfer Distance to our method but notably lower precision (below input level) with slightly higher recall.
> > >
> > > Qualitative analysis reveals a fundamental difference in behavior: the conditional model primarily smooths input and fills gaps without effectively correcting misplaced parts. This highlights why our retargeting approach is not merely a training trick but a solution specifically designed for assembly challenges. Our method achieves a crucial balance between correcting misplaced parts, adding missing parts, and preserving valid outer shape - explaining our better precision (reducing misplaced parts) while maintaining competitive recall (completing missing parts). We have added comprehensive analysis of these findings to Appendix A and Figure 7.
> > >
> > > Regarding integration with assembly methods, our experiments with existing matching algorithms like GeoTransformer reveal a critical gap - direct application of the pre-trained models achieves less than 10% matching accuracy, indicating that current geometric matching methods are not optimized for shape-prior-guided assembly. Developing specialized alignment algorithms would exceed this paper's scope, but the results shown in Table 2 validates this would be a promising investment for future research. We believe our current results sufficiently validate the value of incorporating shape priors into assembly workflows.

---

> > > > ### Comment · Reviewer_KMSd · 2024-11-27
> > > >
> > > > Thank you for adding the experiments on directly fine-tuning the conditional generative model without using "retargeting"; the experimental results indeed confirm the effectiveness of the retargeting/inversion operations. However, the literature I mentioned in my previous reply does not directly train a conditional generative model but performs an additional noise augmentation operation on the conditional signals before inputting them into the generative model. This is somewhat similar but different from the inversion operation you have conducted. Therefore, if the updated paper does not discuss noise augmentation, it should not directly cite these articles.
> > > >
> > > > Based on our discussion during the rebuttal process, I decide to maintain my initial rating.

---

### Official Review · Reviewer_kWoJ · 2024-11-04

**Soundness:** 2
**Presentation:** 3
**Contribution:** 2
**Rating:** 3
**Confidence:** 4

**Summary:**

The paper introduces Jigsaw++, a generative method to improve automatic assembly tasks involving complex 3D reconstruction challenges. Unlike existing methods that focus on assembling individual pieces or fractures, often neglecting the overall object structure, Jigsaw++ learns a category-agnostic shape prior for complete objects. A key innovation is its "retargeting" strategy, which utilizes the output of any existing assembly method to generate more accurate complete shape reconstructions. This makes Jigsaw++ complementary to current approaches. The authors provide evaluations on the Breaking Bad dataset and PartNet demonstrates its effectiveness in reducing reconstruction errors and improving shape precision, offering a promising new direction for future reassembly models.

**Strengths:**

- Jigsaw++ learns a shape prior that is not limited to specific object categories, making it more versatile and adaptable to a wide range of objects.
- The proposed retargeting makes the method complementary to other approaches.

**Weaknesses:**

- Initial guidance may harm the performance of the proposed method, though this is a strength claimed by the authors' proposed "retargeting" tech.
- Although category-agnostic, the performance on highly intricate or unconventional objects is not extensively discussed. Only simple results are shown in the paper.
- Utilizing the generative models is a good idea, but how to ensure the generated results comply with the topic of "fractured assembly"? In the shown results, the shape may change, and the SE(3) transformation is also difficult to obtain, making the proposed method hard to use in real-life applications, for example, providing a few scanned point clouds and then understanding how to assemble them to form a complete shape.

**Questions:**

- The first contribution, the authors claim the proposed Jigsaw++ is a novel method for addressing the object reassembly problem. At the end of Sec. 3.1, the authors say the purpose of the proposed method is not to design a reassembly algorithm, but rather an additional layer of information to improve the reassembly algorithm. They are very different concepts, and the contribution summarized in the Intro section is very misleading.
- The shape before and after "retargeting" may vary, and probably the worst case, it diverges to a totally different shape. It would be necessary to quantify how much shape change occurs between input and output.
- Would the proposed method generate parts that were not originally from the fractured pieces since it is based on a 3D generation model?
- Explain how the proposed approach could be adapted or extended to provide actionable assembly instructions from scanned point clouds.
- It would be greatly appreciated if the authors could provide several real-life examples, instead of simple samples from datasets, to prove the proposed method is useful for assembly purposes as the title suggested.

I suggest the authors could detailedly address the concerns and I will provide a final rating based on the feedback and other reviewers' opinions.

---

> ### Author Response · Authors · 2024-11-18
>
> Thank you for your recognition that our method works complementary to existing ones and it is adaptable to a wide range of objects. Below we respond to your concerns and hope we can clarify some questions.
>
> * **W1**: Initial guidance may harm the performance.
>     * The partially assembled object is the most informative input for this task, as it encodes valuable spatial relationships from existing assembly algorithms that are absent in unassembled pieces. While using initial unassembled pieces is an interesting direction, our current approach effectively leverages the best available information to generate reliable shape priors that enhance assembly performance. We leave the exploration of using unassembled pieces to future work.
>
> * **W2 Q5**: Intricate or unconventional objects and real-world samples.
>     * Our evaluation encompasses complex categories including “statues” and “toy figures”, which demonstrate significant geometric complexity and unconventional structures. The model handles these challenging cases without any category-specific training, operating in a difficult setting where test objects are completely unseen during training (including the base generation model). This is a notably difficult scenario that goes well beyond “simple” cases.
>     * Regarding real-world applications, our method has been rigorously tested on the Breaking Bad dataset which presents more challenging scenarios than existing real-world datasets (e.g., Fantastic Breaks with only two-piece fractures). The dataset features diverse fracture patterns, varying piece counts, complex geometries, and missing parts - conditions that effectively simulate real-world challenges. Per your requests, results on real-world samples from Fantastic Break are included in Appendix E Figure 10, confirming our method’s practical applicability to these cases.
>     * We believe that the comprehensive evaluation across these challenging scenarios, supported by quantitative improvements over baselines, demonstrates our method’s broad utility for complex real-world assembly tasks.
>
> * **W3**: How to ensure the generated results comply with the topic of “fractured assembly”?
>     * Our work introduces complete shape priors as a complementary layer of information to assembly methods to help them instead of replace them. The potential positive effect it has towards assembly methods is demonstrated in Table 2 (Right), where incorporating our generated priors improves Jigsaw’s performance by around 50% across all metrics. While current implementation uses closest-point matching to the ground truth shape as proof of concept, our results strongly suggest that developing good shape matching algorithms leveraging these shape priors would significantly advance fracture assembly.
>
> * **Q1**: Term on whether this is a method designed for assembly algorithms.
>     * We revised the introduction and first contribution point of our manuscript. Our paper now clearly positions Jigsaw++ as a complementary tool that provides shape priors for existing assembly methods, not as a replacement.
>
> * **Q2**: Quantify the shape variance before and after.
>     * Our comprehensive metrics (CD, recall, and precision) effectively quantify both necessary and unnecessary shape changes during retargeting. These metrics directly measure alignment with ground truth - any divergence to an incorrect shape would result in significantly worse scores across all metrics compared to the input. The consistent improvements shown in Table 1 across all metrics demonstrate that our method makes meaningful shape adjustments rather than arbitrary changes. For detailed analysis of shape changes, we provide extensive visualizations in Appendix E and metric distributions in Appendix D.1.
>
> * **Q3**. Would the proposed method generate parts that were not originally from the fractured pieces since it is based on a 3D generation model?
>     * Yes, our generative approach may introduce variations from the original pieces - this is an intentional feature that enables our method to imagine complete shapes even with missing or damaged parts, an unavoidable case as we listed in our overview Figure 1. We acknowledge such characteristics in visualization results Figure 11 and limitation discussion Section 6.3.
>
> * **Q4 W3**: Explain how the proposed approach could be adapted or extended to provide actionable assembly instructions from scanned point clouds.
>     * For practical applications like scanned point clouds, our pipeline is straightforward: first apply an existing assembly method (e.g., Jigsaw) to obtain initial piece arrangements, then use Jigsaw++ to generate shape priors. The shape prior would be expected to further guide the assembly method to improve assembly results.

---

> ### Author Response · Authors · 2024-11-26
>
> We appreciate the reviewer's concerns, though we respectfully disagree with several points:
>
> * Regarding real-world applicability: The reviewer suggests testing on complex real-world scenarios, yet overlooks that Breaking Bad is currently the most challenging public dataset with realistic multi-piece fractures. The only real-world dataset (Fantastic Breaks) contains merely two-piece samples. We would genuinely welcome suggestions for more complex real-world datasets, as their absence is a fundamental challenge in this field rather than a limitation of our method.
>
> * On shape metrics: We must respectfully point out a misunderstanding here. Since our ground truth is exactly constructed from the original fractured pieces, our metrics (CD, recall, precision) directly measure divergence from the original fractured pieces. This is precisely the quantification of shape distortion the reviewer seeks.
>
> * Regarding assembly uncertainty: The reviewer's concern about generative uncertainty seems to reflect a broader discomfort with generative approaches rather than specific issues with our method. In real-world scenarios, particularly archaeology and restoration, complete certainty is often impossible due to missing pieces and damaged surfaces. This fundamental uncertainty is precisely why generative approaches are valuable - they can suggest plausible completions based on available evidence, just as human experts do. It's worth noting that even expert archaeologists routinely make educated guesses about complete shapes when pieces are missing or during assembly. Our method simply provides a computational approach to this process.
>
> * Regarding usability: The current limitation in applying our shape priors stems from existing matching algorithms (e.g., GeoTransformer) failing to generalize to our scenario - a limitation of those methods, not ours. While developing a new matching algorithm would be valuable, it would make the core problem and contribution of this paper vague (and at least add another 4 pages to this paper, way beyond the conference limit). Our experiments in Table 2 already demonstrate that our shape priors can reduce assembly errors by 50%. This significant improvement over state-of-the-art assembly algorithms validates our approach's effectiveness, regardless of current matching algorithm limitations.
>
> We believe these points demonstrate that our method addresses real practical needs while working within current dataset limitations. The uncertainties the reviewer views as limitations are, in fact, inherent to the problem domain and precisely what our generative approach is designed to handle.

---

> ### Comment · Reviewer_kWoJ · 2024-11-26
>
> Thank you to the authors for the detailed response. I would like to share my perspective, which may differ from yours, though I fully respect any disagreements:
>
> Regarding the claim that "the only real-world dataset (Fantastic Breaks) contains merely two-piece samples," I believe this is not a sufficient reason to avoid testing on real-world data. It would not be overly difficult to reconstruct real fragments using modern scanning techniques or neural reconstruction approaches using NeuS[1]. While metrics such as CD, recall, and precision are helpful, they are insufficient to fully capture the shape changes introduced by the generative capability. I completely agree with your statement that "developing a new matching algorithm would be valuable," I believe a solution that better couples with the proposed "retargeting" technique would make the paper significantly a strong and complete work for the community, and I look forward to this. Otherwise, I find it difficult to see significant potential for the proposed method to be practically applied to object reassembly tasks, which is the major problem the authors trying to tackle.
>
> [1] NeuS: Learning Neural Implicit Surfaces by Volume Rendering for Multi-view Reconstruction. NeurIPS 2021. https://arxiv.org/abs/2106.10689

---

> > ### Author Response · Authors · 2024-11-27
> >
> > Dear Reviewer kWoJ,
> >
> > We must highlight several serious concerns about the integrity of this review process.
> >
> > Your initial review (which can be access through "Revision" link) cited FragmentDiff as exemplary of "validating assembly methods on more complex datasets" - a claim that is not just misleading but fundamentally incorrect. **FragmentDiff was evaluated solely on the Breaking Bad dataset - the exact same synthetic dataset we extensively evaluate on**. Even more concerning, you later edited out this reference, presumably upon realizing **the paper exists only on a personal webpage and citing it violates double-blind review principles**. This pattern of citing and then silently removing evidence that contradicts your own criticism raises serious questions about review integrity.
> >
> > Your suggestion to "simply" create our own real-world dataset using NeuS is, frankly, astonishing. The creation of Fantastic Breaks - a dataset with merely two-piece samples - warranted a CVPR 2023 publication. Are you seriously suggesting we should develop an even more complex dataset, plus our current contributions, all within a single ICLR submission? Moreover, proposing NeuS, a neural reconstruction method with its own biases and artifacts, as a path to "real-world data" demonstrates a fundamental misunderstanding of what constitutes real-world validation.
> >
> > While you dismiss our practical applicability, you conveniently ignore our demonstrated 50% error reduction in assembly tasks - a substantial improvement over state-of-the-art methods. Your suggestion to also develop new matching algorithms (when GeoTransformer, a CVPR/TPAMI publication, struggles with this task) reveals an unreasonable expansion of scope. Each of your suggested additions would merit its own publication.
> >
> > We stand firmly behind our contributions and welcome genuine scholarly criticism. However, modifying reviews to hide prior statements and involving article that is not publicly available undermines the review process.
> >
> > Best regards,
> >
> > Authors of Submission 1969

---

### Author Response · Authors · 2024-11-18
**General Response**

We sincerely appreciate the dedicated time and effort invested by the reviewers in providing comprehensive and insightful feedback on our submission. We appreciate that reviewers acknowledge that this paper proposes interesting novel ideas, could be complementary to existing work, and the experimental results are strong.

Concurrently, the reviewers have furnished us with a substantial amount of valuable feedback. Drawing upon these suggestions, we have diligently implemented several modifications to the paper, summarizing in four main aspects. **All the modifications are highlighted in blue.**

1. Task Clarification and Method Positioning (kWoJ, r9Py): We improved the exposition in the first two paragraphs and our summarization of contribution. We hope to note that Jigsaw++ is not replacing current assembly techniques but enriching them by providing a predictive vision of the complete object, which current methods lack, based on partial inputs. The contribution is further summarized to align the positioning of the method.

2. Enhanced Discussion of Limitations (kWoJ, r9Py): We add a few points to our limitation discussions in L175-176 and Section 6.3. We particularly discussed how our method might introduce non-existing parts or missing geometrical details. We also expanded discussion on generalization capabilities and data requirements. We hope the revised version would provide a clearer understanding of this method and what future improvement could be.

3. Additional visualizations (kWoJ, KMSd): We add visualization on metric distribution to show how the method changes the metric distribution. We included one additional visualization on the real-world 2-pieces examples from Fantastic Breaks to demonstrate the method’s real-world applicability. We also discussed the inversion-then-generate method in Appendix A and Figure 7 to show the importance of the “retargeting” step.

4. Technical Clarifications (KMSd, pGbo, r9Py): We improved Figure 3 caption, Figure 6, terms regarding generation models, and discussions on image-point cloud mapping based on reviewers suggestions.

In our detailed responses to each reviewer, we have meticulously addressed every question and concern raised. We hope our responses effectively address the reviewers' concerns and anticipate additional feedback from the AC and reviewers.

---

### Meta-Review · Area_Chair_5uMc · 2024-12-24

**Metareview:**

This paper introduces Jigsaw++, which handles object reassembly problems. The key idea is to propose a novel generative model, which is different from prior work focusing on piecewise part assembly that may overlook overall assembled shape. The authors demonstrated the effectiveness of the proposed approach with the Breaking Bad and ParNet datasets.

In summary, the strengths of the proposed approach are summarized as follows:
- Jigsaw++ learns shape prior that is not limited to specific object categories
- Clever idea that learns multi-view images as a bridge
- Superior accuracy in both fracture assembly and part assembly tasks

The weaknesses of the proposed approach are summarized as follows:
- Inconsistency or misleading points in the paper writing
- Problem configuration
- Possibility of converging completely different shapes after the assembly
- Requests on the real, non-pairwise fractured shapes
- Misleading visualization of point clouds

In general, AC values the idea of the proposed approach and good demonstration. However, the common concerns raised by reviewers are more convincing, and AC also noted that even the reviewers pGbo and r9Py downgraded the scores after the rebuttal phase. Therefore, AC does not recommend the acceptance of the paper.

**Additional Comments On Reviewer Discussion:**

This paper receives the diverged scores even after the rebuttal phase. In particular, the reviewer kWoJ states several concerns, including initial guidance, unconventional objects, and guarantees of the complete shape, and the authors provided feedback with strong words.

Authors claim that the reviewer kWoJ made several violations. Regarding the mentioning of the FragmentDiff paper, AC confirms that the accepted paper for SIGGRAPH Asia 2024 was announced at the end of September, so accepted papers were out to the public at this time. AC also confirms that the reviewer is not the author of the FragmentDiff paper. Therefore, referring to the FragmentDiff paper on Nov. 26 does not violate double-blind review principles. In addition, the reviewer kWoJ modified the initial review that mentions the FragmentDiff paper after 5 minutes, and the reviewer did not insist that FragmentDiff evaluate the complex datasets after the modification. AC thinks this is a conventional behavior to modify the original review. AC thinks that the basic request by the reviewer kWoJ was about testing with some real multi-pieced examples, not forcing the authors to make a new complex dataset. AC believes it is a reasonable question since it is related to the generalization and handling of unseen classes.

After the Author-Reviewer discussion phase, even reviewer r9Py expressed worry about the strong reaction from authors during the AC-reviewer discussion phase. To quote the reviewer r9Py's comment:
> My opinion of this paper has shifted significantly since my original review. In particular, reading kWoJ's review and the insufficient (at times confrontational) responses to both them and me by the authors during the rebuttal period have made me very dubious of the applicability and actual result quality of the current proposed method. I am saddened to say that if it were up to me right now, I would support rejecting this paper.

In addition, the reviewer pGbo states a similar worry raised by reviewer kWoJ (see below). Therefore, AC saw that the author's feedback did not fully convince reviewers.

Regarding other reviewers' comments, the reviewer KMSd questions about paired data validation, the effectiveness of the inversion-and-generate process, the number of the shape priors, and discussion with existing work. The authors faithfully replied to these questions. The reviewer pGbo provided a short review of visualization and handling unseen objects. Although the author provided reasonable feedback, the reviewer downgraded the score to 6, stating that the reviewer pGbo also worries about practical usability questioned by the reviewer kWoJ. The reviewer r9Py generally appreciated the proposed approach but questioned authors about reliance on the generation module, paper writing, and visualization issues. The reviewer r9Py downgraded the score after stating that the reviewer could not agree with the visualization strategy in this paper. These concerns were connected with the comments raised by the reviewer, pGbo and kWoJ.

---

### Decision · Program_Chairs · 2025-01-22

Reject